# A hidden cysteine in Fis1 targeted to prevent excessive mitochondrial fission and dysfunction under oxidative stress

Suman Pokhrel [1,2,6], Gwangbeom Heo [1,6], Irimpan Mathews [3], Shun Yokoi [2,4,5], Tsutomu Matsui [3], Ayori Mitsutake [5], Soichi Wakatsuki [2,4] & Daria Mochly-Rosen [1]

Fis1-mediated mitochondrial localization of Drp1 and excessive mitochondrial fission occur in human pathologies associated with oxidative stress. However, it is not known how Fis1 detects oxidative stress and what structural changes in Fis1 enable mitochondrial recruitment of Drp1. We find that conformational change involving α1 helix in Fis1 exposes its only cysteine, Cys41. In the presence of oxidative stress, the exposed Cys41 in activated Fis1 forms a disulfide bridge and the Fis1 covalent homodimers cause increased mitochondrial fission through increased Drp1 recruitment to mitochondria. Our discovery of a small molecule, SP11, that binds only to activated Fis1 by engaging Cys41, and data from genetically engineered cell lines lacking Cys41 strongly suggest a role of Fis1 homodimerization in Drp1 recruitment to mitochondria and excessive mitochondrial fission. The structure of activated Fis1-SP11 complex further confirms these insights related to Cys41 being the sensor for oxidative stress. Importantly, SP11 preserves mitochondrial integrity and function in cells during oxidative stress and thus may serve as a candidate molecule for the development of treatment for diseases with underlying Fis1-mediated mitochondrial fragmentation and dysfunction.

Mitochondria are dynamic organelles, and fusion and fission maintain mitochondrial size, morphology and function. Fusion enables replenishment of damaged mitochondrial machinery and maintenance of mitochondrial network whereas fission increases mitochondrial number and enables the removal of damaged mitochondria as a mitochondrial quality control measure[1]. A tetratricopeptide repeat (TPR) motif containing mitochondrial outer membrane protein, mitochondrial fission protein 1 (Fis1), mediates mitochondrial fission by recruiting large cytosolic GTPase, Drp1, to the mitochondrial outer membrane[2,3]. In yeast, Fis1 is the only mitochondrial adapter responsible for mitochondrial recruitment of Dnm1, human Drp1 homolog in yeast, to cause mitochondrial fission[4,5]. However, in mammals, it is the

mitochondrial fission factor, Mff, that mediates Drp1 recruitment and physiological mitochondrial fission[2,6,7], whereas Fis1-Drp1 interaction mediates pathological mitochondrial fission[8,9]. Fis1 interaction with Drp1 occurs during cellular stress, but not under physiological conditions, suggesting that cellular signals during stress activate Fis1.

How is stress detected by Fis1 and what structural changes activate Fis1 thus enabling its binding of Drp1 on the mitochondria? Despite being a small and structured protein, the understanding of the structure activity relationship of different regions of Fis1 protein is limited. The cytosolic domain of Fis1 is composed of 6 tight α-helices with two anti-parallel TPR motifs forming a concave hydrophobic surface[3]. Such large hydrophobic patches are critical for protein-

[1]Department of Chemical and Systems Biology, Stanford University School of Medicine, Stanford, CA, USA. [2]Biological Sciences Division, SLAC National Accelerator Laboratory, Menlo Park, CA, USA. [3]Stanford Synchrotron Radiation Lightsource, Menlo Park, CA, USA. [4]Department of Structural Biology, Stanford University School of Medicine, Stanford, CA, USA. [5]Department of Physics, School of Science and Technology, Meiji University, Kanagawa, Japan. [6]These authors contributed equally: Suman Pokhrel, Gwangbeom Heo. ✉e-mail: soichi.wakatsuki@stanford.edu; mochly@stanford.edu

protein interactions[10]. One way to regulate the interactions of such proteins is by autoinhibition, in which part of the protein folds in and occludes the hydrophobic surface, which then folds away in response to certain signals rendering the protein competent for interactions. In yeast, the N-terminal region of Fis1 was shown to be autoinhibitory as it folds inward to occupy the concave hydrophobic surface critical for interaction with Drp1; removal of the first 16 residues in N-terminus increases Drp1 binding[11]. Similarly, a peptide, identified by phage display, binds human Fis1 only after the N terminal α1-helix is removed[12]. However, the N-terminal α1-helix was previously shown to be indispensable for Fis1-mediated mitochondrial fission in mammalian cells[13,14]. This divergence in the function of the N-terminal region of yeast and human Fis1 despite their structural similarity indicates potential additional regulatory role of the N-terminal region in human Fis1. The structural basis of human Fis1 activation, subsequent Drp1 recruitment to mitochondria and regulation of mitochondrial fission under cellular stress is unknown. Recently, Fis1 was found to be phosphorylated at Thr34 during cellular stress by DNA PK[15,16] or at Tyr38 in cancer cells by Met kinase[17] and these phosphorylations increase Drp1 recruitment and cause excessive mitochondrial fission. These studies also show that phosphorylation mimic-mutants, Thr34Asp and Tyr38Glu, faithfully recapitulate the effects of Fis1 phosphorylation in terms of increased Drp1 mitochondrial localization and mitochondrial fission. But how do these modifications 'activate' Fis1 to recruit Drp1?

Here, we study the structural consequences of Fis1 phosphorylation to determine the mechanism of Fis1 activation and to guide us in identifying inhibitors of this Fis1 activation. We show that Fis1 phosphorylation causes conformational change allowing Cys41 mediated Fis1 covalent homodimerization and genetic or chemical abrogation of Cys41 in Fis1 prevents Drp1 localization to mitochondria and excessive mitochondrial fragmentation and dysfunction in cells under oxidative stress.

## Results

### The N-terminal region of Fis1 is dynamic

To investigate the dynamic nature of the N-terminal region of Fis1, we performed molecular dynamics (MD) simulations of wildtype (WT) Fis1. Data obtained from 30 independent 1.2 μs-MD simulations of WT Fis1 were consistent with recently reported findings[18]; we found that the first 10 amino acids of the N-terminus are extremely flexible with average root mean square fluctuations (RMSF) of backbone atoms (Cα, N and C) for these residues ranging from 2.3 to 13.6 Å (Fig. 1a). Despite being structured, residues of α1 helix (aa 11–26) also showed high RMSF values (1.2–2.3 Å), compared to residues that are part of the other helices in the protein (Fig. 1a). Interestingly, the backbone atoms of the residues making the turn between α1-α2 helices (aa 27–31) also showed larger fluctuations (1.2–2.2 Å) compared to the rest of the protein suggesting the potential mobility of the entire α1 helix under perturbations (Fig. 1a). This high fluctuation of backbone of α1 helix suggests that perturbations such as phosphorylation events in this or neighboring helix could have large implications on the integrity of α1 helix.

We next determined the effect of Fis1 phosphorylation at Thr34 or Tyr38 on the dynamics of the N-terminus. These phosphorylations

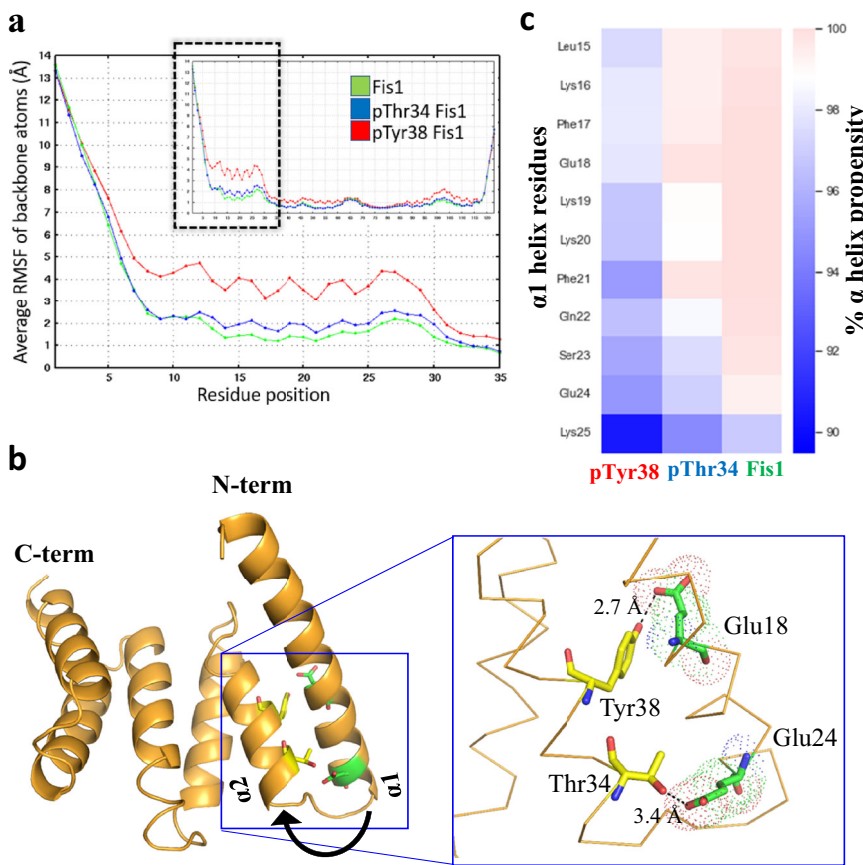

**Fig. 1 | Molecular dynamic (MD) simulations predict dynamic nature of Fis1 N-terminus. a** 30 independent 1.2 μs-MD simulations of Fis1 (green), phosphorylated pThr34 Fis1 (blue) and pTyr38 (red) are shown. Plotted values are mass average root mean square fluctuations of backbone (Cα, C and N) atoms. RMSF of backbone atoms of first 35 residues are highlighted. **b** Relative positions and distances of known phosphorylation sites (Thr34 and Tyr38) and negatively charged residues (Glu18 and Glu24) are shown in yellow and green, respectively. **c** Average structural propensities of residues in the α1 helix residues 15–25 to form α helix are calculated from the MD simulations and shown as heat maps.

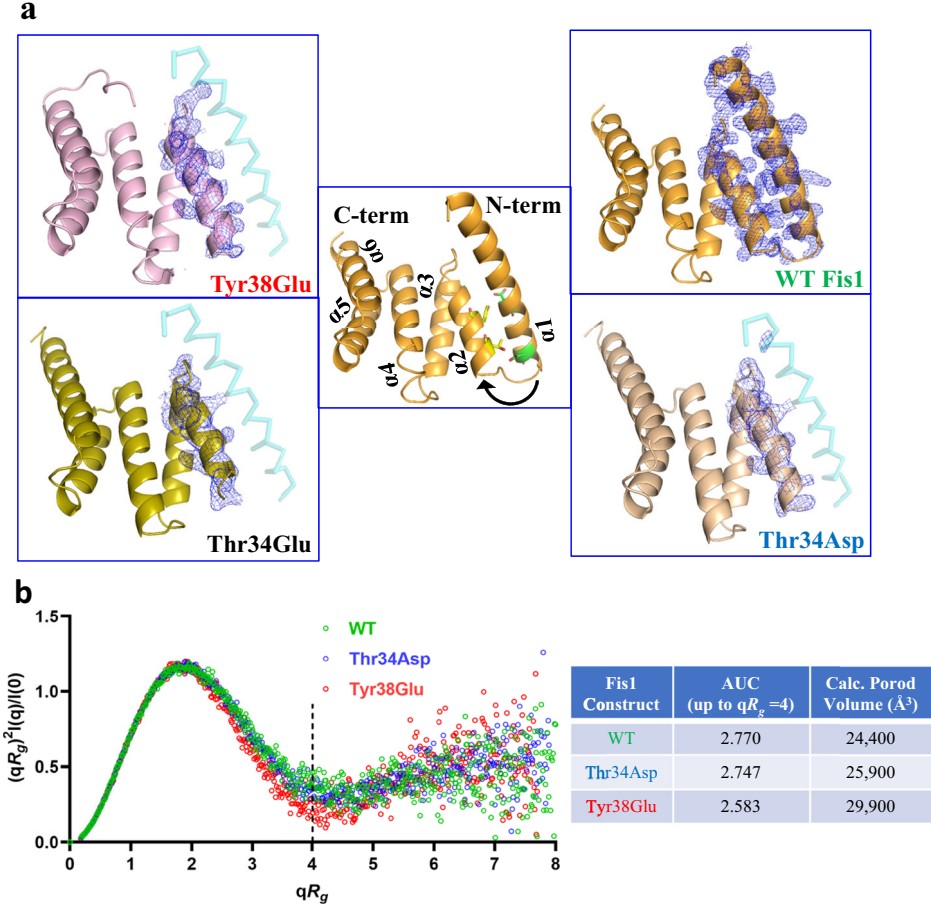

**Fig. 2 | N-terminus undergoes significant conformational change in activated Fis1. a** X-ray structures showing electron density [composite omit map ($\sigma = 1.0$)] of structures of WT Fis1, Thr34Asp Fis1, Thr34Glu Fis1 and Tyr38Glu Fis1. All the mutant structures are missing electron density for α1 helix, which is shown as transparent cyan wire. **b** Kratky plots of monomers of WT, Thr34Asp and Tyr38Glu Fis1 obtained from small angle X-ray scattering data are overlaid and Porod volumes and AUC (up to the $qR_g = 4$) calculated from the SAXS experimental data are modeled. Source data are provided as a Source Data file.

| Fis1 Construct | AUC (up to $qR_g = 4$) | Calc. Porod Volume ($Å^3$) |
|---|---|---|
| WT | 2.770 | 24,400 |
| Thr34Asp | 2.747 | 25,900 |
| Tyr38Glu | 2.583 | 29,900 |

were shown to increase Fis1-mediated Drp1 recruitment to mitochondria and increased mitochondrial fission[15–17]. Thr34 and Tyr38 are on α2 helix in close proximity to Glu24 and Glu18 on the α1 helix, respectively, and the carboxylic acid side chains of these residues are even closer and face each other at a distance of 3.4 Å and 2.7 Å, respectively (Fig. 1b). Based on the proximity of the phosphorylation sites on the α2 helix and the negatively charged residues on the α1 helix, we hypothesized that phosphorylation of Thr34 or Tyr38 will introduce significant charge repulsions between the phosphate group and the carboxylic acid side chain of Glu24 or Glu18, thus making the N-terminus even more flexible. This prediction was supported by 30 independent 1.2 μs-MD simulations of phosphorylated forms of Fis1, Thr34 or Tyr38 (Fig. 1a). The effect of Tyr38 phosphorylation was much greater than that induced by Thr34 phosphorylation; the RMSF values for α1 helix (aa 11–26) and turn between α1-α2 helices (aa 27–31) were higher for the phosphorylated forms (Fig. 1a). We also noticed that the N-terminus of the phosphorylated forms of Fis1 showed instances of α1 helix unfolding (Fig. 1c).

## The α1 helix of Fis1 undergoes conformational change upon activation

Using MD simulations, we observed instances of disruption of the integrity of the α1 helix, particularly in the phosphorylated forms of Fis1. We calculated the average propensity of residues in α1 helix (aa 15–25) to form α-helical structure using the DSSP algorithm in 30 independent 1.2 μs-MD simulations for WT, pThr34, and pTyr38. Compared to unphosphorylated Fis1 (WT), the propensity of amino

acid residues 15–25 to form α-helical structure in pThr34 and pTyr38 were lower by 0.19–2.22% and 1.91–6.44%, respectively (Fig. 1c, Supplementary Fig. 1, Supplementary Movie 1). We also performed the φ and ψ angle analysis in these simulations. We found that there were differences in standard deviations, but the differences in the average φ and ψ values were smaller (Supplementary Table 1). These MD simulations data indicate the potential fluctuation and conformational change in the α1 helix, which could enable the exposure of a concave hydrophobic surface in Fis1 for protein-protein interactions.

To explore this possibility further, we crystallized and determined the X-ray structures of the WT and three phosphorylation mimic-mutants of Fis1: Thr34Asp Fis1, Thr34Glu Fis1 and Tyr38Glu Fis1 (Fig. 2a). Each of these mutants was previously shown to be sufficient to increase Drp1 recruitment to mitochondria and subsequent mitochondrial fission in cell-based studies[15–17]. We postulated that these mutants should inform us about the structural changes that activate Fis1. Consistently, we observed that the first 30 residues including the α1 helix were missing from the structure providing strong evidence that the α1 helix undergoes significant conformational change and is potentially disordered in the phosphorylated forms, whereas it was intact in the WT (Fig. 2a). In one of the instances, Tyr38Glu Fis1 mutant was crystallized in an intermediate state, where the α1-helix was still intact, but was shifted away from the rest of the protein, as compared to the WT (Supplementary Fig. 2). This observation further corroborates the results from the MD simulations and the dynamic nature of the N-terminus of Fis1.

To confirm that these experimental observations are not crystallographic artifacts, and that similar differences in the structures can be

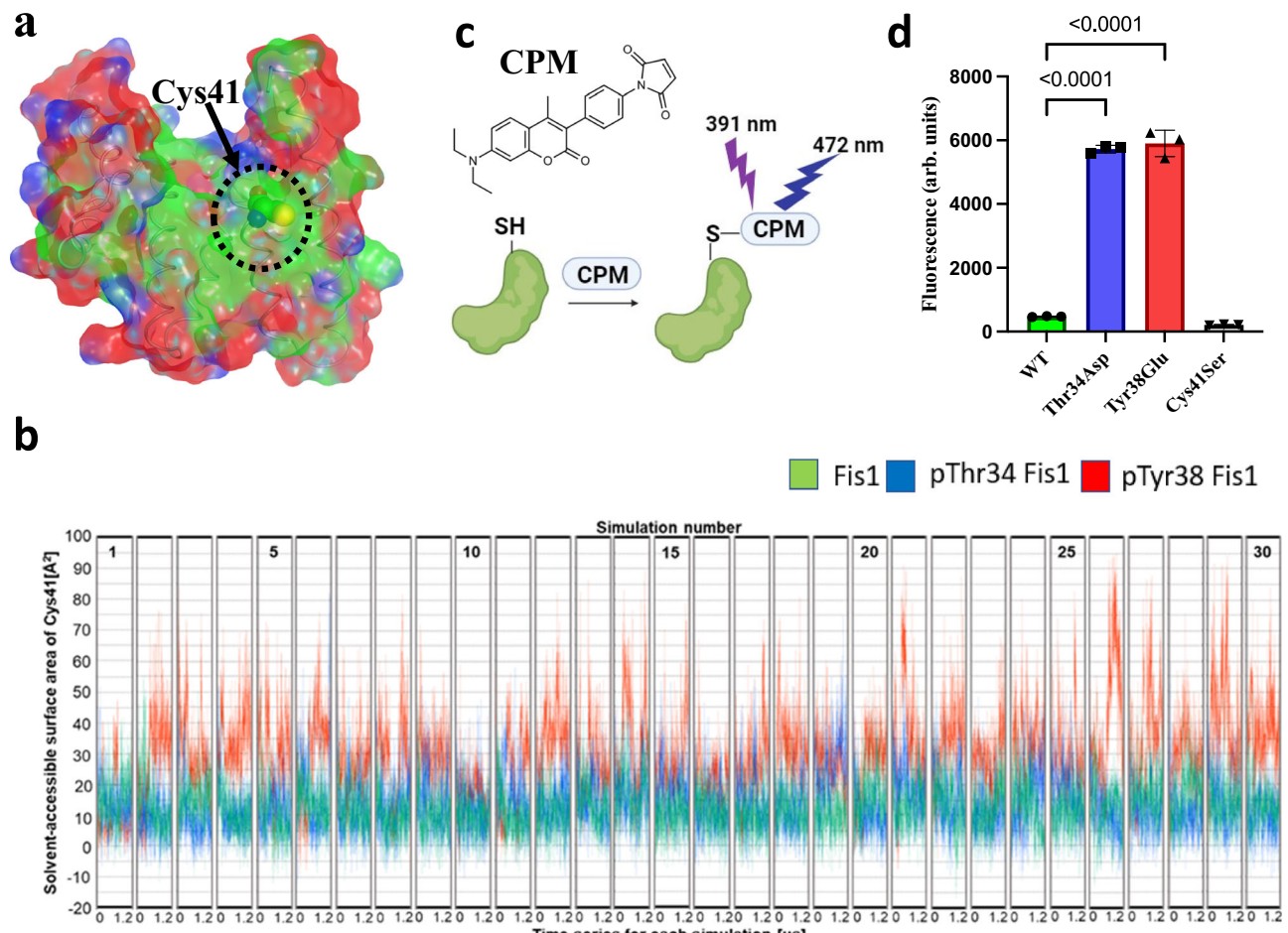

**Fig. 3 | Conformational change in N-terminus exposes Cys41. a** Fis1 protein shown with a surface and the Cys41 residue is highlighted. The surface is colored according to sequence conservation in vertebrates. Residues conserved in all vertebrates (except birds) are shown in green and residues that are not conserved are shown in red. Sequences aligned for this analysis are provided in the Supplementary Information file (Supplementary Fig. 20). **b** Solvent accessible surface area of Cys41 (obtained from molecular dynamics simulations) for Fis1 (green), phosphorylated Thr34 Fis1 (pThr34; blue) and phosphorylated Tyr38 Fis1 (pTyr38; red) are shown. Each box represents an independent 1.2 μs simulation. The average SASAs of Cys41 in WT, pThr34, and pTyr38 calculated from the 30 independent 1.2

μs MD simulations each were $12.5 \pm 8.5$ Å$^2$, $13.9 \pm 10.0$ Å$^2$, and $29.2 \pm 14.7$ Å$^2$, respectively. **c** Schematic of the assay used to measure Cys41 exposure using maleimide based fluorescent probe, CPM. Created in BioRender. Pokhrel, S. (2025) https://BioRender.com/xpiwgmn. **d** CPM fluorescence-based binding assay is shown. 4 μM protein and 8 μM CPM were used. Plotted value represents the mean of 3 independent experiments each with 5 technical replicates. Error bar represents standard deviation. One-way ANOVA with Dunnett's multiple comparisons test (Thr34Asp and Tyr38Glu each against WT) was performed (p-values <0.0001). Source data including all statistics (degrees of freedom, p-values, effect sizes, and confidence intervals) are provided in the Source Data file.

observed in solution, we used size exclusion chromatography coupled with small angle X-ray scattering (Fig. 2b, Supplementary Table 2, Supplementary Fig. 3). In solution, Thr34Asp Fis1 and Tyr38Glu Fis1 had different structures compared to the WT Fis1 (Fig. 2b). We then hypothesized that if the α1 helix undergoes conformational change and moves away from the rest of the protein, then the structures of the Fis1 phosphorylation mimic mutants should appear larger compared to WT Fis1. Indeed, based on the SAXS data, the Porod volumes for the WT, Thr34Asp and Tyr38Glu Fis1 were 24,400Å$^3$, 25,900Å$^3$ and 29,900Å$^3$, respectively. These data strongly support our hypothesis, and the trend is consistent with our observation in the MD simulations, where Tyr38Glu Fis1 had the biggest difference, followed by Thr34Asp Fis1 and WT Fis1 (Fig. 1a, c).

## Conformational change in α1-helix and Fis1 activation expose the one and only cysteine, Cys41, in Fis1

Human Fis1 protein contains a single cysteine, Cys41, in the entire protein. Cys41 and a stretch of hydrophobic amino acids around this residue is conserved in vertebrates suggesting a critical functional role

(Fig. 3a). A previous study showed that Fis1 forms a Cys41-mediated covalent dimer in vitro, but the formation of the dimer was minimal, and the propensity of dimerization increased after unfolding and refolding[19]. These data suggest that Cys41 is not exposed in the basal state but can be exposed and can mediate dimerization upon conformational change and subsequent solvent exposure. Additionally, Fis1 has been shown to be activated through phosphorylation under oxidative stress induced by cisplatin[15], lipopolysaccharide (LPS)[15], rotenone[15] or hydrogen peroxide[16] treatments in cells in culture and oxidizable cysteine residues are well known to act as redox sensors[20]. Hence, we hypothesized that this Cys41 residue could work in a similar way, conformational changes in N-terminus would expose the thiol group of Cys41, thus making it available for oxidation.

We performed all-atom MD simulations of unphosphorylated Fis1, pThr34 and pTyr38 and calculated the solvent-accessible surface area (SASA) for Cys41 based on the LCPO method[21]. Cys41 in both pThr34 and pTyr38 were more exposed to the solvent than that in the WT-Fis1, as expected (Fig. 3b). The average SASAs of the Cys41 in the unphosphorylated Fis1, pThr34 and pTyr38 during the 1.2 μs × 30 MD

simulations were $12.5 \pm 8.5$ Å$^2$, $13.9 \pm 10.0$ Å$^2$ and $29.2 \pm 14.7$ Å$^2$, respectively (Fig. 3b). These data indicate that Cys41 in pThr34 and pTyr38 are more exposed to the solvent than in the unphosphorylated Fis1. This trend of Cys41 exposure is consistent with the trend seen in N-terminus fluctuation, propensity of conformational change in α1-helix, and the SAXS data, where pTyr38 has the greatest effect, followed by pThr34 and unphosphorylated Fis1.

To corroborate these findings from the MD simulations, we investigated biochemically whether the Cys41 residue in the phosphorylation mimic-mutants, Thr34Asp Fis1 and Tyr38Glu Fis1, is more exposed compared to the WT Fis1. To this end, we used a thiol reactive maleimide-based fluorescent probe, CPM (7-diethylamino-3-(4-maleimidylphenyl)-4-methylcoumarin), to measure the cysteine reactivity in solution (Fig. 3c). We first established that all the fluorescence signal in our system is coming from CPM reaction with the thiol group of Cys41 by using a Cys41Ser Fis1 mutant, which has no thiol groups. Co-incubation of Cys41Ser Fis1 and CPM probe produced very low fluorescence signal (Fig. 3d). WT Fis1 showed minimal fluorescence signal, but the phosphorylation mimic-mutants had approximately over 12-fold higher fluorescence signal, suggesting that the cysteine is more exposed in the phosphorylation-mimic mutants compared to the WT (Fig. 3d). These data also support observations from molecular dynamics simulations and SAXS, where the N-terminal region was found to be dynamic, and it underwent structural change and moved away from the concave hydrophobic surface, thus exposing the normally hidden Cys41 under conditions that mimic its activation. There was a difference in Cys41 exposure in MD simulations data in the phosphorylated forms of Thr34 and Tyr38 but there was no major difference in the amount of chemical modification after 30 min incubation of these protein constructs with CPM. It is possible that we could have observed differences if the CPM fluorescence measurements were performed at shorter time points.

## Cys41 can be oxidized, when exposed, to mediate covalent Fis1-Fis1 homodimerization

As mentioned previously, human Fis1 protein was reported to have propensity to form small amount of covalent dimers, mediated by Cys41[19]. At the time of their publication, it was difficult to rationalize this observation because of the unexposed position of Cys41 residue, buried in between α1-α2 helices, thus making it unavailable for disulfide bridge formation with another Fis1 protomer (Supplementary Fig. 4). However, in light of the evidence that the α1 helix can undergo significant conformational change and move away from the concave hydrophobic surface, it is conceivable that Cys41 can form a disulfide bridge and hence mediate Fis1 covalent homodimer. We crystallized the phosphorylation-mimic mutants, Thr34Asp, Thr34Glu and Tyr38-Glu Fis1, and we observed that in all the cases, these mutants crystallized as covalent dimers mediated by Cys41 disulfide bridge between two Fis1 protomers (Fig. 4a, Supplementary Fig. 5a, b). As Thr34Asp, Thr34Glu and Tyr38Glu Fis1 have a greater propensity for a conformationally altered N-terminus (Fig. 1 and Fig. 2) and hence have more exposed Cys41, they crystallized as covalent dimers.

In cells, Fis1 is localized to the mitochondrial outer membrane via a transmembrane anchoring sequence, but all the in vitro data presented in this study used only the cytosolic domain of Fis1 protein. To overcome this limitation, we rationalized that tethering the cytosolic domain of Fis1 to His-tag affinity magnetic beads could better resemble the biological context of Fis1 by restricting the degrees of freedom when Fis1 is tethered to a membrane. When the cytosolic domain of Fis1 was tethered to His-tag affinity magnetic beads, the propensity of phosphorylation mimic constructs to form dimer increased under non-reducing non-heat denatured condition (Fig. 4b). This dimer could be reversed to monomer in reducing condition using 2-mercaptoethanol (2-ME), suggesting that Fis1 homodimer formation was mediated by Cys41 (Fig. 4b). The propensity of dimerization of immobilized phosphorylation mimic Fis1 constructs increased with time - with more dimer than monomer at longer time points while the WT Fis1 construct yielded no dimer. (Supplementary Fig. 6).

Upon closer examination of the crystal structures, we observed that the concave hydrophobic surface in the covalent dimers of all the phosphorylation-mimic mutants was exposed (Fig. 4c). Such exposed concave hydrophobic surfaces were reported to be critical for scaffolding proteins to mediate homo-oligomerization or interaction with other protein partners[10].

Unlike phosphorylation mimic mutants, the WT Fis1 crystallized as a non-covalent dimer. A similar non-covalent Fis1 dimer was also reported previously[3]. We rationalized that the α1-helix and N-terminus structure are intact in WT Fis1, making Cys41 unavailable for disulfide bridge formation and hence WT Fis1 could not form a covalent dimer (Supplementary Fig. 4). However, due to the inherent dynamic nature of N-terminus of Fis1 as seen in MD simulations and CPM fluorescence assay (Figs. 1a, 3d), WT Fis1 showed some propensity to form covalent dimers in solution as reported previously[19]. We observed that the concave hydrophobic surface that is known to mediate protein-protein interactions in TPR motif-containing proteins is between the interface of the dimer in both our current and the previously reported structure (Fig. 4d). However, in the active form, the concave hydrophobic surface should be exposed to mediate homo-oligomerization or interaction with other proteins; yeast Fis1 interaction with Caf4, bacterial secretion system protein PscG interaction with PscE both require concave hydrophobic surfaces (not sufficient in the case of Fis1/Caf4)[22–24]. We therefore hypothesized that the WT dimer observed by X-ray crystallography was an artifact due to crystal packing and the dimer of Fis1 should be different in solution. Indeed, the dimer structure determined from crystallography did not fit the SAXS curve (Fig. 4e), suggesting that the Fis1 dimer in solution is structurally different than that observed in the crystal. When we modeled the structure of Fis1 WT dimer using the Thr34Asp Fis1 covalent dimer with modeled the missing unstructured first 30 residues, the Fis1 covalent WT dimer perfectly fitted the SAXS curve for WT Fis1 dimer (Fig. 4f). We generated the ab initio density and envelopes using the experimental SAXS curve, implementing both P1 and P2 symmetry. We did not observe significant differences in the shape of the envelope when either P1 or P2 symmetry was enforced. The modeled WT Fis1 covalent dimer fitted both the P1 and P2 symmetry enforced ab initio envelops independently calculated from the experimental SAXS curve (Fig. 4g). This evidence indicates that the Fis1 dimer that is present in solution is not related to the WT crystallographic dimer; these data combined with the data from the non-reducing non-heat denatured SDS-PAGE gel (Fig. 4b) suggest that the small amount of Fis1 dimer in solution is indeed a covalent dimer. However, it is also possible that the differences of WT Fis1 dimers in solution and crystals may represent different types of dimers that enable interactions with different protein partners.

## Structural insights guide assay design for screening specific inhibitor of activated Fis1

We posited that since Cys41 is exposed when Thr34 or Tyr38 is phosphorylated, and these phosphorylations mediate Fis1 activation and subsequent pathological mitochondrial fission and dysfunction[15–17], compounds that bind to Cys41 should inhibit Fis1 function to induce a protective effect during oxidative stress. We used the structural insights from the X-ray crystallography, SAXS and MD simulations to strategize our discovery effort for small molecule that can specifically inhibit activated Fis1 without interfering with the Fis1 in the basal state with intact α1-helix structure.

Above, we confirmed that Cys41 is exposed only in the activated Fis1 but is occluded in the basal state (Fig. 3d). Therefore, small molecules that bind to Fis1 by tightly engaging Cys41 should be specific for the activated Fis1 as this residue is unavailable in the Fis1 in basal

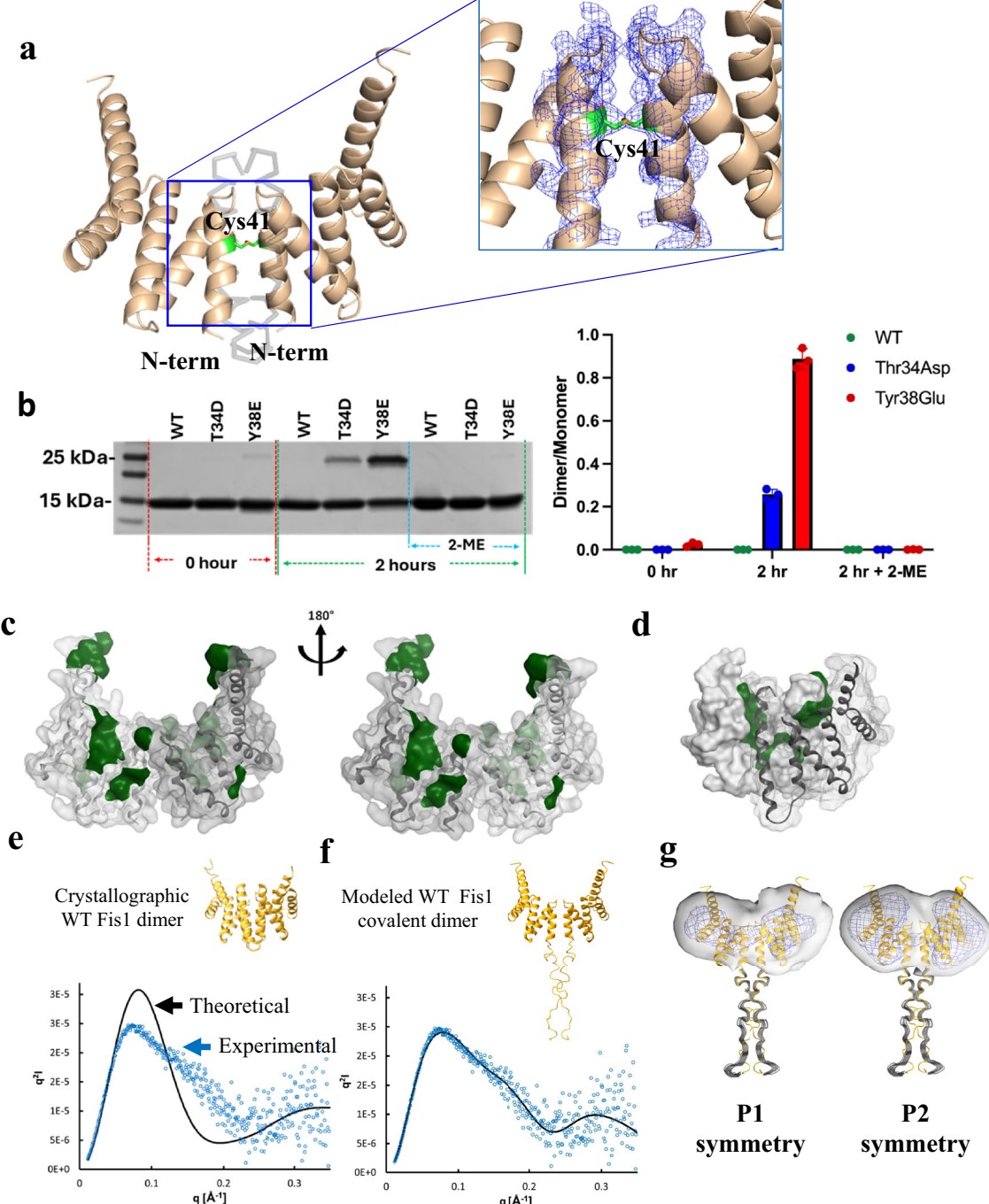

**Fig. 4 | Cys41 is oxidizable and mediates covalent dimerization. a** X-ray structures showing electron density [composite omit map (σ = 1.0)] for Thr34Asp Fis1 covalent dimer. Missing N-terminal residues AA 1–32 is shown as grey transparent wire. Continuous electron density for disulfide bridge between the Cys41 of two monomers is highlighted. **b** Non-reducing non-heat denatured SDS-PAGE gel showing covalent dimers for different constructs of Fis1 [pThr34 (T34D), pTyr38 (Y38E) and wildtype (WT)] under different conditions; equal amount of beads initially loaded with 20 μg protein (30% of total initial beads; about 6 μg protein) were loaded in each lane. Mean of dimer/monomer band ratios from three independent experiments are plotted. Error bars represent S.D. **c** Thr34Asp Fis1 covalent dimers have large hydrophobic patches exposed (green). The structure is rotated by 180° to show the patches on both sides. **d** Crystal structure of WT Fis1 dimer has the concave hydrophobic patches buried between the interface of the dimer. **e** The theoretical Kratky plot of WT crystallographic dimer structure doesn't fit the experimental Kratky plot of WT Fis1 dimer. **f** The theoretical Kratky plot of the WT Fis1 covalent dimer modeled based on Thr34Asp Fis1 dimer structure with modeled N-terminal region fits the experimental Kratky plot of WT Fis1 dimer. The theoretical plots are shown in solid black line and experimental plot is shown in blue circles in **e** and **f. g** The modeled WT Fis1 covalent dimer (in yellow ribbon) also fits both the P1 and P2 symmetry enforced ab initio envelops (white envelope and dark blue mesh at 2 contour levels, 0.13 and 0.057) independently calculated from experimental SAXS curve. Source data are provided as a Source Data file.

state. We predicted that a small molecule with cysteine reactive warheads would be an ideal candidate for this purpose. We first used the CPM probe, a fluorescently labelled thiol reactive maleimide, as proof of concept to see if it binds to Cys41 in activated Fis1 and if this binding

can be protected by incubating the protein with unlabeled maleimide (0–50 μM) (Fig. 5a). We used phosphorylation mimic-mutants of Fis1, Thr34Asp and Tyr38Glu, with exposed Cys41 as models for activated Fis1. These mutants produced a robust fluorescence signal with CPM

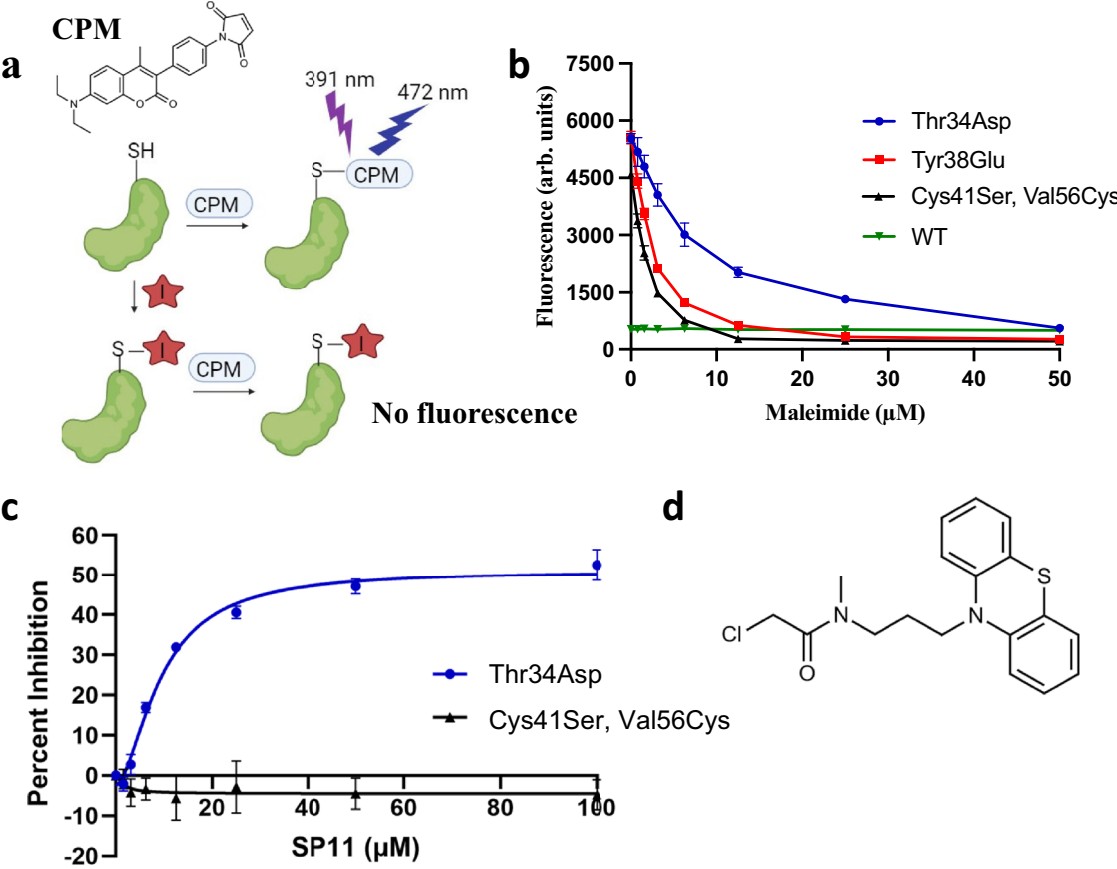

**Fig. 5 | Structural insights guided assay development and discovery of activated Fis1 inhibitor. a** Schematic of CPM assay for the screening of covalent inhibitor of activated Fis1. Created in BioRender. Pokhrel, S. (2025) https://BioRender.com/fyg7416. **b** Proof of concept of CPM-based screening assay using maleimide. Unlabeled maleimide (0-50 μM) inhibited CPM fluorescence in Thr34Asp Fis1 and Tyr38Glu Fis1. 4 μM protein and 8 μM CPM were used in the assay. Each data point represents mean of 3 independent experiments each with 3 technical replicates. Error bar represents standard deviation. **c** Inhibition of CPM fluorescence by SP11 in Thr34Asp Fis1 but not in counter screen construct (Cys41Ser, Val56Cys Fis1 double mutant). 4 μM protein and 8 μM CPM were used in the assay. Each data point represents mean of 3 independent experiments each with 3 technical replicates. Error bar represents standard deviation. Source data, including all statistics are provided in the Source Data file. **d** Structure of SP11.

and this signal was inhibited by preincubating the protein with unlabeled maleimide (Fig. 5b), suggesting the suitability of the assay to screen for covalent inhibitors that can bind to Cys41 only in activated Fis1. We used Thr34Asp Fis1 as a model for activated Fis1 in the high-throughput screen and preincubated it with cysteine-reactive small molecules followed by incubation with CPM. When the small molecules were covalently bound to Cys41, fluorescence signal of thiol-bound CPM decreased as thiol group of Cys41 was already occupied; this reduction in fluorescence signal was used as a readout of Cys41 binding (Fig. 5a).

To avoid molecules with excessively strong electrophilic warheads, like the maleimide, which can bind to any protein with cysteines due to intrinsic reactivity to thiol, we used a secondary screen with the Cys41Ser Fis1 in which we introduced a new cysteine residue on the surface, Val56Cys on α 3-helix. We validated that this construct has a similar fluorescence profile when treated with CPM and that the fluorescence can be inhibited with unlabeled maleimide like Thr34Asp or Tyr38Glu Fis1 (Fig. 5b). We then screened 6000 molecules from Enamine (Kyiv, Ukraine) with thiol reactive covalent warheads in a 384 well format to identify molecules that bind to Thr34Asp Fis1 and not the counter screen, Cys41Ser Val56Cys Fis1 double mutant, and discovered one hit, SP11. SP11 is a molecule with phenothiazine core and methyl-substituted chloroacetamide warhead, SP11 that bound to Cys41 of Thr34Asp Fis1 and not to Cys56 of Cys41Ser, Val56Cys Fis1 double mutant counter screen (Fig. 5c, d).

## SP11 covalently binds specifically to Cys41 of activated Fis1

SP11 inhibited CPM fluorescence of Thr34Asp Fis1 with an apparent IC50 of 9.4 μM (Fig. 5c). As the binding of CPM dye to Thr34Asp Fis1 is 1:1, we had to use an extremely high concentration of Fis1, 4 μM, to achieve a robust fluorescence signal and good separation of signal to noise. SP11 binding to Thr34Asp Fis1 is also 1:1 and therefore the apparent IC50 of 9.4 μM translates to about 2.5 molecules of SP11 for every half molecule of Thr34Asp Fis1, suggesting that SP11 is a potent inhibitor. SP11 did not bind to the counter screen Cys41Ser, Val56Cys Fis1 double mutant even at a high concentration (Fig. 5c). Inhibition by SP11 plateaued at about 55% inhibition, which may reflect solubility issue at the higher concentrations. It may also reflect the dynamic nature of the α1-helix and N-terminus; not all the Thr34Asp Fis1 proteins have homogenous SP11 binding pocket or Cys41 exposure. Indeed, incubation of Thr34Asp Fis1 with SP11 for a longer time period increased percent inhibition of CPM fluorescence and approached almost 100% at 52 hours (Supplementary Fig. 7). As expected, preincubation of Fis1 with SP11 inhibited the dimerization of immobilized Thr34Asp Fis1 (Supplementary Fig. 8).

We next used MS/MS-based peptide mapping and found that the Fis1-derived peptides are labeled by SP11 at Cys41, leading to a mass shift by 310 Da (Fig. 6a, Supplementary Fig. 9). To confirm that the inhibition of Fis1 dimerization with SP11 is not an artifact of the covalent warhead, we tested an analogue, SP22, with the same phenothiazine core but with acrylamide warhead (Supplementary Fig. 10a). Similar to SP11, SP22 also bound to Cys41 of Thr34Asp Fis1 and not to

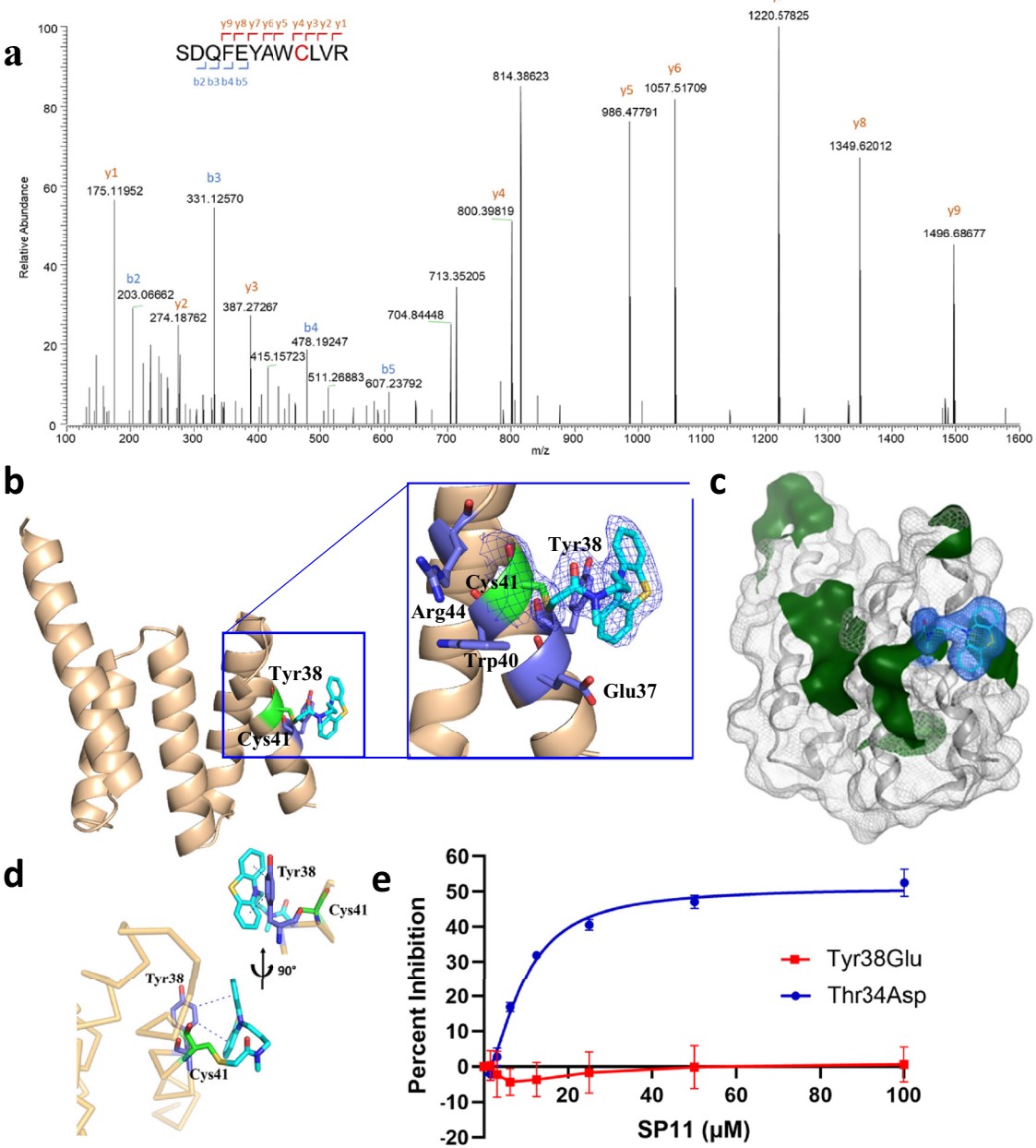

**Fig. 6 | SP11 binds to Cys41 of activated Fis1. a** MS-MS spectrum showing Fis1 peptides are modified at Cys41 by SP11. **b** X-ray structures showing SP11 (cyan) bound to Thr34Asp Fis1. Electron density [2Fo-Fc map (σ = 1.0)] for SP11 and pocket residues (shown as blue sticks) are highlighted. **c** Phenothiazine moiety sits on the hydrophobic patch formed by Tyr38. **d** Two benzene rings of SP11 (cyan) form CH-Pi interactions with Tyr38 residue (blue). **e** CPM fluorescence inhibition assay for

SP11. SP11 binds to Cys41 in Thr34Asp Fis1 mutant but does not bind to Cys41 when Tyr38 is mutated to Glutamic acid. 4 μM protein and 8 μM CPM were used in the assay. Each data point represents mean of 3 independent experiments each with 3 technical replicates. Error bar represents standard deviation. Source data are provided as a Source Data file.

Cys56 of Cys41Ser, Val56Cys Fis1 double mutant counter screen, but with slightly higher apparent IC50 of 15.4 μM (Supplementary Fig. 10b, c). This data with SP22 establishes that the binding of these molecules to Thr34Asp Fis1 is driven by the binding of phenothiazine core to the binding pocket of Thr34Asp Fis1 and not just due to the intrinsic reactivity of the chloroacetamide/acrylamide warhead.

**Co-crystal structure of Thr34Asp Fis1 with SP11**
Co-crystal structure of Thr34Asp with SP11 was solved at 1.9 Å resolution (Fig. 6b). Consistent with the previous Thr34Asp Fis1 structure, the first 30 residues in the N-terminus were also missing in this structure (Figs. 2 and 6b). A clear and continuous density for SP11 arising from Cys41 residue was also observed,

indicative of a covalent bond between Cys41 and SP11 (Fig. 6b). Besides the covalent bond, the two benzene rings of the phenothiazine core of SP11 made two CH-Pi interactions with Tyr38 (Fig. 6c, d). Other interactions of SP11 with Fis1 are exclusively hydrophobic and the phenothiazine core of SP11 assumes a slightly bent conformation to complement with the topology of the hydrophobic surface formed by Tyr38 (Fig. 6d).

To validate the significance of the CH-Pi interaction with Tyr38 in the crystal structure, we mutated the Tyr38 residue in Fis1 to a Glu residue, which we previously showed also has exposed Cys41 and was compatible with CPM probe-based fluorescence assay (Figs. 3d, 5b). We saw that SP11 did not bind to the Tyr38Glu Fis1 even at 100 μM, suggesting that interaction with Tyr38 is required for SP11 binding,

validating the crystal structure (Fig. 6d, e). SP22 also did not bind to Fis1 with Tyr38Glu mutation (Supplementary Fig. 10d).

## SP11 inhibits Fis1-mediated pathological mitochondrial fission and dysfunction during oxidative stress

We tested the effect of SP11 in Fis1-mediated mitochondrial fission and mitochondrial reactive oxygen species (ROS) production during oxidative stress using HK-2 cells in culture. Previously, Thr34 residue was shown to be phosphorylated in HK-2 under cellular stress[15]. Using MitoSOX, we found that 250 nM of SP11 inhibited mitochondrial ROS generation when cells were treated with 50 μM hydrogen peroxide compared to the DMSO-only treatment (Fig. 7a). Cells treated with 100 μM hydrogen peroxide showed mitochondrial fragmentation, and this fragmentation was also prevented by SP11 treatment (Fig. 7b–d). There was a drastic reduction in the average mitochondrion size per cell in hydrogen peroxide-treated cells and this was prevented by SP11 treatment (Fig. 7c). We saw that more than half of the cells showed fragmented mitochondria when treated with hydrogen peroxide but only 8% of the cells had fragmented mitochondria in control. 250 nM of SP11 treatment reduced the fraction of cells with fragmented mitochondria from 52% in hydrogen peroxide treated cells back to 8%, similar to control cells (Fig. 7d). SP11 treatment also increased the fraction of cells with fused mitochondria by about 50% compared to hydrogen peroxide-only-treated cells. We also determined the amounts of mitochondria-associated Drp1 in HK-2 cells in the presence of 100 μM hydrogen peroxide. Peroxide-induced Drp1 mitochondrial localization was inhibited by treatment with 250 nM of SP11 (Fig. 7e). Role of the Cys41 residue of Fis1 was further confirmed using Cys41Ser knock-in homozygous Be(2)-M17 cells. The mutant cells with Cys41Ser Fis1 showed less sensitivity to 50 μM hydrogen peroxide treatment in the MitoSOX assay and treatment with SP11 had no effect in these cells, whereas SP11 treatment significantly inhibited the mitochondrial ROS in WT cells treated with hydrogen peroxide. (Fig. 7f). SP11 did not show any effect in non-stimulated cells (Supplementary Fig. 11). As Be(2)-M17 cells are not suitable for mitochondrial imaging since the cells are small with a very small cytosolic (non-nuclear) area, we studied the mitochondrial morphology of Fis1 KO MEFs transfected with WT and Cys41Ser Fis1 expressing plasmids in different treatment conditions (Fig. 7g, Supplementary Fig. 12). In Fis1 KO MEFs transfected with plasmid expressing WT Fis1, 50 μM hydrogen peroxide treatment caused mitochondrial fragmentation. Almost 50% of the cells showed a fragmented mitochondrial phenotype in the presence of hydrogen peroxide and pretreatment with 250 nM SP11 significantly inhibited mitochondrial fragmentation. Whereas Fis1 KO MEFs transfected with a plasmid expressing Cys41Ser Fis1 didn't show fragmented mitochondrial phenotype in the presence of hydrogen peroxide and pretreatment with SP11 had no effect. We also measured Drp1 mitochondrial translocation in these Fis1 KO MEFs transfected with a plasmid expressing WT and Cys41Ser Fis1. We observed that Drp1 translocation to mitochondria was significantly increased upon hydrogen peroxide treatment in the cells with WT Fis1, but mitochondria associated Drp1 level was unchanged in the cells with Cys41Ser Fis1(Supplementary Fig. 13). These data taken together strongly suggest the critical role of Cys41 mediated Fis1 covalent dimerization in Drp1 recruitment and mitochondrial fragmentation and that the effect of SP11 is dependent on Cys41 in Fis1.

## Discussion

How does Fis1 recruit Drp1 to mitochondria only during cellular stress and not under basal conditions? Previous data showed that the N- terminus of Fis1 inhibits it: autoinhibition of Fis1 by the inwardly folded N-terminal region was shown in yeast[11]. More recently, the similar regulatory nature of N-terminal region of human Fis1 was demonstrated using computational, biophysical and functional methods[18,25,26]. These studies showed that N-terminal region of Fis1 is flexible and can adopt different conformations and hence can regulate the activity of Fis1. In this study, we identified how a conformational change in the N-terminus activates Fis1 to bind Drp1. Our crystal structure not only shows the absence of α1 helix but also shows that there is no space for structured α1 helix in the covalent dimer as this space is occupied by α2 helix of another monomer (Supplementary Fig. 4). This observation was consistent across all modeled phosphorylated Fis1 structures and phosphorylation mimic-mutants, using different biochemical, biophysical and structural biology experiments, including MD simulations, small angle X-ray scattering in solution and X-ray crystallography of WT and phosphorylation mimic mutants.

We also report that this conformational change and displacement of the α1-helix activates Fis1 protein and exposes the one and only cysteine residue, Cys41, in Fis1. Oxidizable cysteine residues that perceive oxidative stress and function as redox sensors are found ubiquitously in many proteins[20]. Here we showed that Cys41 is also oxidizable and mediates Fis1 covalent dimerization through disulfide linkage. This oxidation-induced Fis1 dimerization was obvious in the crystal structures of Thr34Asp, Thr34Glu and Tyr38Glu Fis1, where we saw a clear electron density for the disulfide linkage. We also showed that this covalent dimers in Fis1 have concave hydrophobic surfaces exposed, which is conducive for self-interaction or interaction with other proteins. In cells expressing Fis1 in which Cys41 was mutated to Serine, $H_2O_2$-induced Drp1 recruitment to mitochondria, mitochondrial fragmentation and elevation of mitoROS did not occur and treatment with SP11, which binds to Cys41 in WT Fis1, consistently phenocopied the cell expressing Fis1 that lacks Cys41 and Fis1 knock out cells. Although a direct physical interaction between Drp1 and Fis1 dimer was not demonstrated in this in culture study, these results establish that Cys41, which in vitro we showed to mediate Fis1 dimerization, is required for Drp1 translocation to mitochondria under oxidative stress.

We observed higher R-free values for the phosphorylation mimic mutant crystal structures. As R-free values vary depending on several key factors such as resolution and the number of residues that are disordered in the crystal. The latter is particularly important in this case since all the mutants and Fis1-SP11 structures lack α1 helix, i.e., 1/6th of the total amino acids. The atoms of the α1 helix are included in the protein structure but are disordered and not visible. This contributed significantly to high R-free values in the structures reported in our study.

Previously, noncovalent nature of Fis1 dimer was reported in yeast[27] where it is the only mitochondrial adapter that recruits Drp1[4,5]. Whereas in mammals, Fis1 recruits Drp1 only under pathological conditions[8,9]. Therefore, it is not surprising that the effect of dimerization in mammals (exclusively under pathological conditions) and in yeast (under physiological condition) leads to different functional effects on the ability of the protein to recruit Fis1. Cys41 in Fis1 is conserved in vertebrates, suggesting a critical function. In birds, however, even though the stretch of hydrophobic amino acids (36-FEYAW**X**LVRS-45) is conserved, like in other vertebrates, birds have glycine residue instead of Cys41. It is interesting to note that birds have high metabolic rates and have high energy demands and it is likely that cells have higher ROS burden and oxidative stress[28]. Along with other adaptations, such as higher levels of antioxidants e.g., glutathione[29] and fewer mitochondria per cell in birds with longer lifespan[30], this divergence in Fis1 sequence could also be protective against oxidative stress. Additionally, birds are known to be relatively insensitive to LPS even at high concentrations[31], while in other vertebrates LPS has been shown to be one of the stress agents that causes Fis1 phosphorylation

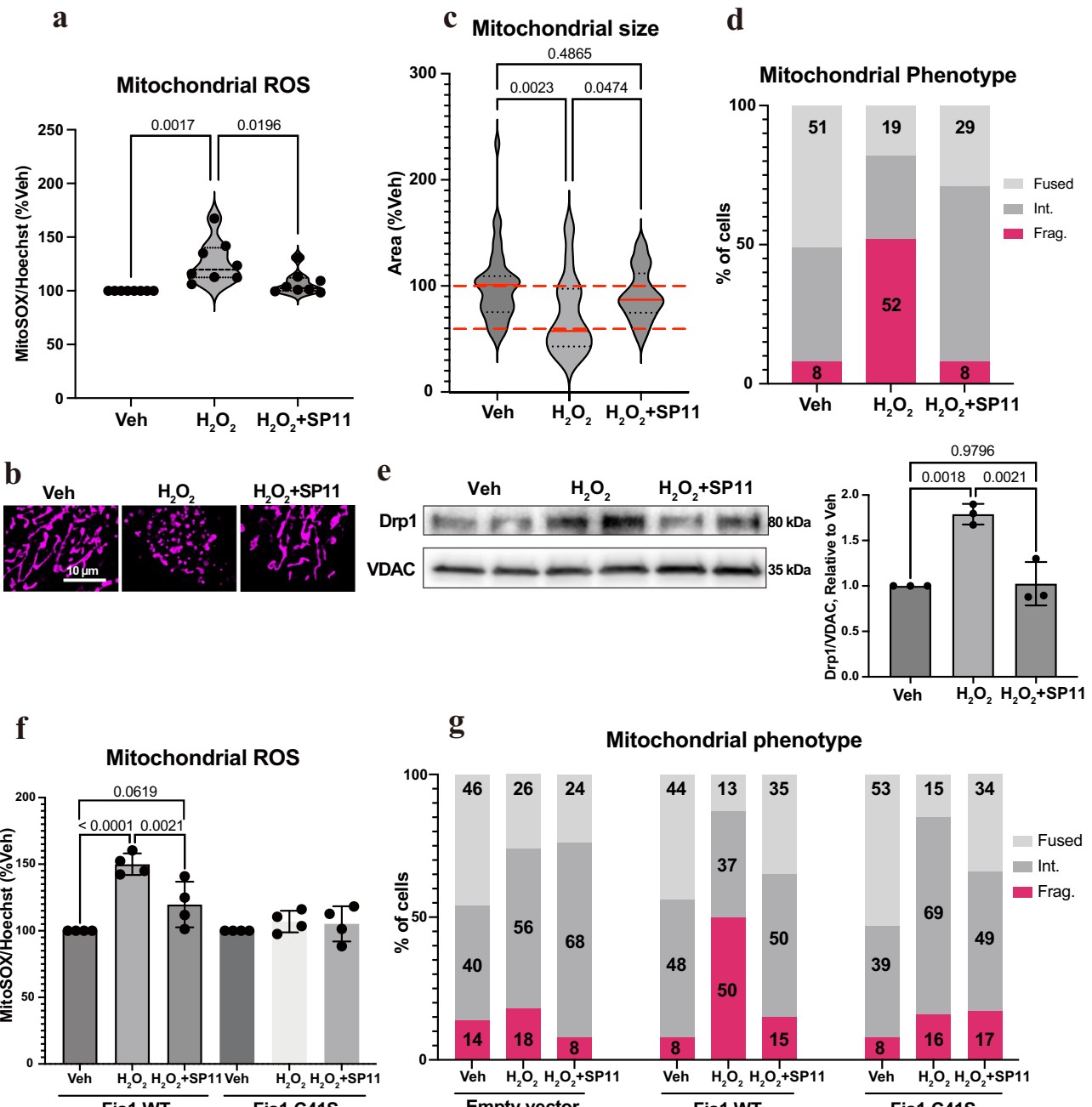

**Fig. 7 | SP11 reduces H₂O₂-induced mitochondrial ROS production and fragmentation and increased Drp1 translocation to the mitochondria. a** HK-2 cells were co-treated with H₂O₂ (50 μM) and SP11 (250 nM) for 24 h. Mitochondrial ROS was measured using MitoSOX™ mitochondrial superoxide indicator. MitoSOX™ signal was normalized to Hoechst 33342. Each data point represents the mean of an independent experiment with 6 replicates each (*n* = 8). **b, c** Mitochondrial morphology in live HK-2 cells was analyzed using MitoTracker Deep Red FM (*n* = 39 cells for Veh; 27 cells for H₂O₂; 38 cells for H₂O₂ + SP11) after co-treatment of H₂O₂ (100 μM) and SP11 (250 nM) for 24 h. Average mitochondrial area per cell was normalized to the vehicle mean. (Center line, median; upper and lower dotted lines, quartiles) **d** Cells from (**c**) were categorized into three groups with two cut-off criteria, 100% and 60% of the Veh group. **e** HK-2 cells were pre-treated with SP11 (250 nM) or DMSO vehicle control for 30 min. Then the cells were treated with H₂O₂

(100 μM) in the presence of SP11 or DMSO, and incubated for additional 24 h. Densitometry graph was obtained from three independent experiments. Data are presented as mean values +/− S.D. **f** Be(2)-M17 WT and Cys41Ser cells were co-treated with H₂O₂ (50 μM) and SP11 (250 nM) for 24 h. Each data point represents the mean of an independent experiment with 6 replicates each (*n* = 4). **g** Fis1 knock-out MEFs were transfected with empty vector, Fis1 WT or Fis1 C41S plasmid for 24 h. The cells were pre-treated with vehicle control or SP11 (250 nM) for 30 min, then treated with vehicle control or H₂O₂ (50 μM) for additional 6 h. Mitochondrial morphology in live cells was analyzed using the same method as HK-2 cells. The original violin plot is presented in Supplementary Fig. 12. One-way ANOVA with Tukey's multiple comparisons test was performed. Veh vehicle Int. intermediate Frag. fragmented, WT wild-type. Error bar represents S.D. Source data are provided as a Source Data file.

at Thr34 position and causes excessive mitochondrial fragmentation and dysfunction[15].

Small molecule inhibitors for small, membrane-bound, scaffolding proteins have rarely been described previously; such proteins that

mediate protein-protein interaction are considered 'undruggable', partly because of the solvent-exposed inconspicuous "pockets"[32]. Also, the lack of enzymatic assays for structural proteins makes it difficult to screen for small-molecule binders or inhibitors. In this study, using

structural insights, we strategized an assay method to find specific inhibitor of activated Fis1, using maleimide-based fluorescent probe, CPM. We measured the fluorescence signal as a result of direct covalent binding of CPM to Cys41 and the decrease of this signal due to small molecule binding served as a readout for small molecule binding. Using this strategy, we identified a first-in-class covalent inhibitor that binds to activated Fis1. SP11 inhibits Thr34Asp Fis1 CPM fluorescence in vitro with an apparent IC50 of 9.4 μM. As SP11 binding to Thr34Asp Fis1 is 1:1 and irreversible, this apparent IC50 is underestimated, as we used 4 μM Thr34Asp Fis1. This half inhibitory concentration of 9.4 μM at 4 μM Thr34Asp Fis1 effectively translates to 5 molecules of SP11 binding 1 molecule of Fis1, indicating excellent potency. However, we were unable to measure $K_i$ and determine Kinact/$K_i$ for SP11 as Fis1 is a structural protein without any known enzymatic activity; there are no direct in vitro functional assays to measure Fis1 inhibition. We also co-crystallized SP11 with activated Fis1 and showed that it binds covalently to Cys41 residue, and the phenothiazine moiety of SP11 adopted a very slightly bent conformation to make two CH-Pi interactions with Tyr38 residue. We also validated our crystal structure using site-directed mutagenesis where we observe no SP11 binding when Tyr38 residue is mutated. Importantly, we showed that 250 nM SP11 inhibits Fis1-mediated mitochondrial localization of Drp1, mitochondrial fission and dysfunction induced by oxidative stress in culture, supporting the efficacy of SP11.

SP11 contains a phenothiazine moiety with a chloroacetamide warhead connected by a three-carbon chain. Phenothiazine has been extensively studied as pharmacologically active molecule and several molecules with phenothiazine substitutions have been already approved by the FDA for different indications including nausea, vomiting, allergy and neurological diseases[33]. This scaffold is known to have polypharmacology by engaging many targets[34]. The addition of covalent warhead to phenothiazine in SP11 may reduce its engagement with other targets. With fragments of smaller size, specificity is always an issue and one rarely gets away without elaborate medicinal chemistry to achieve selectivity. However, polypharmacology may also be beneficial for SP11 in certain indications; phenothiazines by themselves have been shown to have beneficial effect in certain neurodegenerative diseases[35] and SP11 with its covalent warhead can disrupt pathological Fis1-Drp1 interaction and promote mitochondrial integrity and health in nervous tissues, as we have shown before for other inhibitors of Drp1/Fis1 binding[36–38]. Many phenothiazine-containing drugs are orally bioavailable, have longer half-lives and are blood-brain barrier permeable[35,39] and thus, it is likely that SP11 has good drug-like properties. SP11 is an irreversible inhibitor and also has excellent potency and therefore should have a beneficial effect even at lower dose, minimizing potential toxicological liabilities. SP11 includes a Cys reactive moiety and therefore there is a formal possibility that $H_2O_2$-induced mitochondrial fission is inhibited by interaction of SP11 with Cys-containing proteins other than Fis1. However, SP11 does not significantly affect mitochondrial fragmentation or ROS production in cells expressing Cys41Ser Fis1. Importantly, our data shows that binding of SP11 to any other Cys (if any) in a Cys41Ser homozygous Fis1 cell is insufficient to reduce mitochondrial fragmentation or ROS production. SP11 does not bind Cys41Ser Fis1 nor does it bind Fis1 Cys41Ser Val56Cys double mutant. Finally, in addition to Cys41, Tyr38 is essential for SP11 binding to Fis1; Tyr38Glu does not bind SP11, indicating further selectivity of the SP11 to its binding site on Fis1. Although Fis1 knockout cells and Cys41Ser Fis1 mutant cells were largely non-responsive to SP11 treatment, we observed slight reduction in percentage of cells with fragmented mitochondria in Fis1 knockout condition and increase in percentage of cells with fused mitochondria in Cys41Ser Fis1 mutant condition upon SP11 treatment. Therefore, it is possible that SP11 with a reactive Cys moiety has some off-target effect. Nevertheless, our findings that

SP11 inhibits $H_2O_2$-induced excessive mitochondrial fission and ROS burden in cells expressing WT Fis1 support our conclusion that SP11 acts primarily by inhibiting Fis1-mediated Drp1 binding.

The role of Fis1 in Drp1 recruitment to mitochondria and excessive mitochondrial fragmentation and dysfunction was previously underappreciated, but recently Fis1-Drp1 interaction has been recognized for its role in driving mitochondrial fragmentation and dysfunction in diseases, such as Parkinson's disease[8], Huntington's disease[36], cardiac ischemia and reperfusion injury[16,40], endotoxemia[15,41,42], cancer metastasis[17] etc. P110[8], a peptide inhibitor of Fis1-Drp1 interaction which we rationally designed, was efficacious in preclinical mouse models of diseases by improving mitochondrial integrity, function and reducing disease-associated pathologies[38,41–43]. These data indicated that disruption of Fis1-Drp1 interaction is a viable and effective strategy to correct dysregulated mitochondrial dynamics to alleviate diseases. Chemical agents, peptides and small molecules known thus far that disrupt Fis1-Drp1 interaction, exert their effect by engaging and reducing GTPase activity of Drp1, which is also involved in Mff-mediated physiological fission and in several other cellular processes. Therefore, inhibitors of Fis1-Drp1 interaction that bind Fis1 are more likely to be safer as therapeutic agents and perhaps more beneficial for chronic diseases, especially those in fast proliferating tissues (e.g., wound healing) and in pediatric diseases with high metabolic demands. Our study focusses on the role of activated Fis1 in causing excessive mitochondrial fission and dysfunction through increased Drp1 recruitment to mitochondria. In addition to promoting fission through Drp1, it is possible that the inhibition of fusion machinery by activated Fis1 also contributes to the fragmented mitochondrial phenotype under cellular stress. Previously, it was shown that Fis1 inhibits proteins involved in mitochondrial fusion such as Mfn1/2 and Opa1[44] and this inhibitory effect of activated Fis1 needs to be investigated.

## Methods

### CPM probe and small molecules

7-Diethylamino-3-(4-maleimidylphenyl)-4-methylcoumarin, CPM probe, was obtained from Abcam (ab145275; Cambridge, UK). Small molecules SP11 and SP22 used in this study were purchased off the shelf initially from Enamine (Kyiv, Ukraine). Independent batches of SP11 and SP22 were also synthesized, characterized and verified. Chemical characterization of the compounds were performed at Enamine (Kyiv, Ukraine) and are provided below and the spectral data is available in the Supplementary Information file (Supplementary Fig. 14, 15).

N-(3-(10H-phenothiazin-10-yl)propyl)-2-chloro-N-methylacetamide (SP11). 1H NMR (DMSO-d6, 500 MHz) δ 7.23−7.18 (2H, m), 7.18−7.11 (2H, m), 7.07 (1H, d), 7.00 (1H, d), 6.94 (2H, dtd, J = 10.76, 7.49, 1.11 Hz), 4.23 (2H, d, J = 68.78 Hz), 3.87 (2H, dt, J = 26.19, 6.87 Hz), 3.44−3.33 (2H, m), 2.82 (3H, d, J = 87.89 Hz), 1.91 (2H, dp, J = 35.03, 6.94 Hz). 13C NMR (DMSO-d6, 151 MHz) δ 166.37, 166.03, 145.11, 128.08, 128.05, 127.69, 127.61, 123.18, 123.01, 116.61, 116.28, 46.94, 44.48, 41.99, 33.60, 25.57. HRMS (m/z): calc. for C18H19ClN2OS (M + ) 346.09057, obs. 346.09806.

N-(3-(10H-phenothiazin-10-yl)propyl)-N-methylacrylamide (SP22). 1H NMR (DMSO-d6, 500 MHz) δ 7.25−7.10 (4H, m), 7.01 (2H, dd, J = 17.73, 8.07 Hz), 6.94 (2H, dt, J = 11.62, 7.42 Hz), 6.59 (1H, ddd, J = 82.58, 16.59, 10.35 Hz), 5.99 (1H, ddd, J = 53.22, 16.65, 2.53 Hz), 5.48 (1H, ddd, J = 131.90, 10.37, 2.54 Hz), 4.02−3.77 (2H, m), 3.55−3.36 (2H, m), 2.87 (3H, d, J = 81.60 Hz), 1.90 (2H, p, J = 6.74 Hz). 13C NMR (DMSO-d6, 126 MHz) δ 164.99, 144.67, 144.63, 127.72, 127.56, 127.17, 127.11, 122.66, 122.50, 116.18, 115.76, 46.09, 45.20, 25.59, 24.45. HRMS (m/z): calc. for C19H20N2OS (M + ) 324.12834, obs. 325.13679.

### Molecular dynamics (MD) simulations

The systems for MD simulations were constructed using the wild-type structure of Fis1 (residue 1-123) revealed by X-ray crystallography

published in this study. The structures of pThr34 and pTyr38 were modeled using the molecular operating environment (MOE; Chemical Computing Group, Montreal, Canada). Protonation states were determined by Adaptive Poisson-Boltzmann Solver (APBS)- PDB2PQR, which is a module solvation force library package with pH 7.5[45]. The structures of WT-Fis1, pThr34, and pTyr38 were solvated in TIP3P[46] water using Amber 18[47,48]. In each system, NaCl was added to neutralize the charge for the Fis1 and to reach a concentration of 150 mM. All MD simulations were performed using Amber 18, with Amber ff14SB and phosaa14SB as the force field for the protein and phosphorylated amino acids, respectively[49]. MD simulations were performed under periodic boundary conditions in a rectangular box. For simulations, the time step was 2 fs, and the trajectory interval was 10 ps. The temperature was maintained at room temperature (298.15 K) using a Langevin thermostat, and the pressure was maintained at 1 bar using a Berendsen barostat. The shake method was also employed. Short-range electrostatic and van der Waals forces were cut off at 1 nm. Firstly, energy minimization was performed under constant volume conditions without pressure control. Then, the 30 independent systems for the WT-Fis1, the pThr34 and the pTyr38 were heated to 298.15 K over 100 ps with harmonic restraint to the protein using $k = 10.0$ kcal/(mol·Å²). Following the heating process, a 5 ns equilibration run for each was conducted without restrain. Finally, 30 independent 1.2 µs-MD simulations were performed for the WT-Fis1, the pThr34 and the pTyr38 (1.2 µs × 90 MD simulations in total), and their production-run data were extracted every 2 ns for analysis. Since we focused on the atomic positional fluctuation of the α1 helix, the root mean square fluctuation (RMSF) of backbone atoms (Cα, N and C) was calculated by fitting to the average structure except for the α1 helix (aa 31-123). The average α-helix propensity for each amino acid residue was calculated based on the DSSP algorithm[50]. The solvent-accessible surface area (SASA) for Cys41 was calculated based on the LCPO method[21]. Overview of the MD simulations is provided in Supplementary Table 3.

## Recombinant Fis1 protein production

Fis1 codon optimized sequence in frame with HRV 3 C protease cleavage site and hexa-histidine tag at the C-terminus was cloned into a pET-28a vector (Genscript, NJ, USA). BL21 (DE3) *E. coli* cells transformed with Fis1 expression plasmid were grown in LB media at 37 °C in a shaking incubator (200 rpm) to an OD600 of 0.6. Cells were then induced with 0.5 mM IPTG (GoldBio, MO, USA) and grown for 16 h at 18 °C. The cells were then collected as pellets by centrifuging the culture for 15 min at 3000×g. All steps after this were performed at 4 °C. E. coli pellet was resuspended in lysis buffer (50 mM Tris pH 8, 150 mM NaCl). Resuspended cells were sonicated for 2.5 m (1 s on and 4 s off) at 70% amplitude using a QSONICA SONICATORS instrument (500 Watts). The whole cell lysate was then centrifuged at 20,442× *g* for 80 min and the soluble fraction was loaded onto a Ni-NTA Agarose (Qiagen, MD, USA) gravity column pre-equilibrated with lysis buffer. The Ni-NTA column was then washed with 50 column volumes of wash buffer (50 mM Tris pH 8, 150 mM NaCl and 40 mM Imidazole). Fis1 protein bound to Ni-NTA was then eluted with elution buffer (50 mM Tris pH 8, 150 mM NaCl and 400 mM Imidazole). Eluted Fis1 protein was then buffer exchanged using Zeba™ Spin Desalting Column, 7 K MWCO, 10 mL (89894; Thermo Fisher Scientific, MA, USA) into lysis buffer (50 mM Tris pH 8, 150 mM NaCl). Buffer exchanged Fis1 was mixed with 6xHis-tagged HRV 3 C protease in 40:1 (W/W), incubated overnight at 4 °C and was passed through Ni-NTA Agarose (Qiagen) gravity column to remove un-cleaved hexa-histidine tagged impurities including the HRV 3 C protease. Hexa-histidine tag free Fis1 was obtained as the flowthrough was flash frozen and stored at −80 °C. This protocol was also followed to express and purify the Fis1 mutants used in this study. Fis1 sequences are provided in the Supplementary Information file (Supplementary Fig. 16).

## CPM fluorescence assay

Recombinant Fis1 protein from above was diluted to 4 µM concentration in PBS and 48 µL of this solution was added to each well in a 96-well black flat bottom chimney plate. 1 µL of test small molecule with cysteine reactive covalent warhead dissolved in DMSO was added to the protein solution to obtain desired final concentration. The test small molecule with cysteine reactive covalent warhead and protein were incubated at room temperature for 30 min. 1 µL of CPM in DMSO to achieve 10 µM final concentration was added to each well with the protein and test small molecule with cysteine reactive covalent warhead. This reaction mixture was incubated at room temperature for 30 min and fluorescence readout (Ex: 391 nm / Em: 472 nm) was taken using Tecan M1000 microplate reader (Tecan, Männedorf, Switzerland). No test small molecule control was taken as 100% signal and the percent inhibition of this signal was calculated at different concentrations of the test small molecule.

## Fis1 immobilization experiment

20 µg of recombinant Fis1 protein in 100 µL lysis buffer (50 mM Tris pH 8.0, 150 mM NaCl and 10 mM 2-ME) was added to 20 µL of HisPur™ Ni-NTA magnetic beads (Thermo Fisher Scientific, USA) and washed three times with wash A buffer (50 mM Tris pH 8.0, 150 mM NaCl, 10 mM 2-ME and 0.1% Tween-20). The mixture was left shaking at room temperature for 1 h. The beads with bound Fis1 protein were separated from the elution using magnetic rack and washed 3 times with wash B buffer (50 mM Tris pH 8.0, 150 mM NaCl and 0.1% Tween-20). Beads with bound protein were resuspended in 100 µL wash B buffer and incubated at room temperature while shaking for time period as indicated. Beads were then separated from the elution using magnetic rack and resuspended in buffer containing 50 mM Tris pH 8.0 and 150 mM NaCl and analyzed using SDS-PAGE (in the presence or absence of 2-ME as indicated). Protein samples were loaded into 4–20% SDS-PAGE gel in the absence of reducing agent and without boiling as described previously[51]. For SP11 treatment, Thr34Asp Fis1 protein was diluted to 4 µM in lysis buffer without 10 mM 2-ME and preincubated with 50 µM SP11 for 1 h at room temperature before adding to the beads. The experiment was performed similarly as above except 2-ME was not added in any buffer. No pre-existing Thr34Asp Fis1 dimer was broken and inhibition of new Fis1 dimer formation by SP11 was monitored (Supplementary Fig. 8).

## Crystallization, data collection and structure determination

Tag-free recombinant Fis1 protein/mutant was concentrated to 8 mg/ml using 3-kDa molecular weight cutoff Amicon Ultra15 ultrafiltration membrane. For Thr34Asp Fis1-SP11 complex crystallization, Thr34Asp-Fis1 was diluted to 4 µM in 50 mM Tris pH 8, 150 mM NaCl buffer and 10-fold molar excess of SP11 was added and incubated for 1 h at room temperature. Thr34Asp Fis1-SP11 complex was then concentrated with a 3-kDa molecular weight cutoff Amicon Ultra15 ultrafiltration membrane to 8 mg/ml.

Each of the protein sample went through screening with around 960 conditions at two different protein concentrations using Oryx 8 crystallization robot in a sitting drop setup. The WT protein was crystallized from screen condition C9 (25% PEG 5KMMe, 0.15 M NaCl, 0.033 M HEPES (pH 7.5), 0.033 M ADA (pH 6.5) and 0.033 M Tris.HCl (pH 8.0)) of a screen developed at SSRL and the crystals cryo-cooled with well solution supplemented with 20% glycerol. The Thr34Asp mutant was crystallized from Pact screen condition E2 (0.2 M sodium bromide, 20% w/v PEG 3350). The crystal was cryo-cooled with well-solution supplemented with 10% glycerol. The Thr34Glu mutant was crystallized from Top96 crystallization solution F2 (0.2 M ammonium acetate, 0.1 M HEPES (pH 7.5) and 25% PEG 3350) and the crystals cryo-cooled with well solution supplemented with 5% glycerol. The Tyr38-Glu mutant was crystallized from Grass2 crystallization solution F11 (0.1 M sodium phosphate dibasic dihydrate, 20% PEG 3350) and the

crystals cryo-cooled with well solution supplemented with 10% glycerol. The Tyr38Glu mutant that showed movement of the N-terminal helix was crystallized from BCS screen condition A1 (0.1 M sodium acetate (pH 4.5) and 30% PEG smear low) and cryo-cooled with well solution supplemented with 3% glycerol. The SP11 covalent complex was crystallized from MemGold crystallization screen G12 (0.001 M zinc sulfate, 0.05 M HEPES, 28% PEG 600) and the crystal was cryo-cooled directly from the drop.

The diffraction data for wild type, Thr34Asp, Thr34Glu, and the two Tyr38Glu mutants were collected at 100 K at the BL12-2 beamline of the Stanford Synchrotron Radiation Lightsource using PILATUS 6 M detectors. The diffraction data for the SP11 covalent complex was collected at the same beamline using the EIGERX 16 M detector. All diffraction data were collected with 360° of data per crystal and 0.2° oscillation per image. For each crystal, diffraction data were merged and processed with XDS[52]. The structures were solved by molecular replacement with MOLREP[53] or PHASER[54] using the coordinates of cytosolic domain of human Fis1 (PDB code: 1NZN) as the search model. Iterative rounds of model-building and refinement were performed with the programs COOT[55] and REFMAC[56]. The final rounds of refinements were performed by REFMAC or PHENIX[57]. The details of data collection and refinement for the higher resolution data are presented in Supplementary Table 4.

## Peptide mapping and MS analysis
Thr34Asp Fis1 was diluted to 1 mg/ml in 50 mM Tris pH 8, 150 mM NaCl buffer and SP11 was added to the protein solution at 100 μM final concentration. The reaction mixture was incubated at room temperature for 1 h and was flash frozen. Peptide mapping and MS analysis was performed at BGI Global Genomic Services (CA, USA).

Samples (25 μg) were denatured with the treatment of 9 M urea followed by the reduction with dithiothreitol (DTT) and alkylation with 2-iodoacetamide (IAM). 10 μL Trypsin/LysC mix stock (0.2 μg/μL) was added to the sample for digestion overnight at 37 °C. The reaction was then quenched with 10% TFA. Digested samples were desalted and eluted by the standard C18 Stage-tip Clean Up protocol. Peptides were then dried by SpeedVac. Samples were reconstituted with $H_2O$ containing 0.1% Formic acid. An aliquot of 10 μL was subjected to the LC-MS/MS system equipped with Thermo Fisher Vanquish LC and Q Exactive HF-X mass spectrometer. MS Data was analyzed with PMI Byos (Protein Metrics, CA, USA) software and sequence provided to reveal peptides coverage on the Fis1 protein.

## SEC-SAXS experiments and analyses
The SEC-SAXS experiments were performed at the Stanford Synchrotron Radiation Lightsource (SSRL) Bio-SAXS beamline 4-2[58]. SEC-SAXS data were collected using a Superdex 75 Increase 3.2/300 column (Cytiva, MA, USA) using SEC running buffer (50 mM Tris pH 8.0, 150 mM NaCl), with samples concentrated with the buffer at 6 or 5 mg/ml for Fis 1 wild-type dimer and others, respectively. 500 images were acquired with 1 s exposure every 5 s at a flow rate of 0.05 ml/min. After the 100th image (blank data collection), the x-ray shutter was closed until just before sample elution to keep the sample cell clean. Data reduction and initial analyses were performed using the BL4-2 automated SEC-SAXS data processing and analysis pipeline, SECPipe (https://www-ssrl.slac.stanford.edu/smb-saxs/node/1860). It implements the programs SASTOOL (https://www-ssrl.slac.stanford.edu/smb-saxs/node/1914) and ATSAS AUTORG[59]. The data were plotted as I(q) versus q, where $q = 4\pi \sin(\theta)/\lambda$, 2θ is the scattering angle, and λ is the wavelength of the X-ray. After careful manual inspection, a total of 5 images were selected to generate the average profile for further analysis. Experimental and analytical details are summarized in Supplementary Table 2.

The program GNOM was used for the indirect Fourier transformation to estimate the distance distribution function P(r)[60]. The theoretical scattering profile of the wild-type dimer was calculated and fitted to the experimental data using the program CRYSOL[61]. SAXS modeling of the wild-type dimer was performed using the program CORAL[59]. A covalent dimer in the Thr34Asp crystal structure (aa: 31–129) was employed as the initial model and the first 30 residues were reconstructed by CORAL. The initial model was fixed during modeling. 20 independent runs with P2 symmetry were performed. The model with the lowest χ2 value was selected as the best model.

An ab initio electron density map of the wild-type dimer was calculated using the program DENSS[62], with a final map calculated by averaging 20 independent runs (the final resolution = 20.75 Å). Ab initio shape determination was also performed by the program DAMMIF[63]. 20 independent models with P2 symmetry were generated. The resulting models were averaged (ICP = 2.46 +/− 1.89) and filtered using the program DAMAVER[64]. Docking of the CORAL model to the SAXS envelopes was performed manually using ChimeraX[65].

## Cell cultures
HK-2 cells were purchased from American Type Culture Collection (VA, USA). The cells were kept in their growth medium DMEM/F12 (SH30023.01; Cytiva) supplemented with 10% fetal bovine serum (FBS; 100-500-500; GeminiBio, CA, USA), 1% penicillin/streptomycin (P/S; 15140-122; Thermo Fisher Scientific), and 5 ng/ml human epidermal growth factor (hEGF; PHG0311; Thermo Fisher Scientific). Wild-type and genetically engineered Be(2)-M17 cells were purchased from Synthego (CA, USA). The cells were kept in growth medium which was 1:1 mixture of MEM (10370-201; Thermo Fisher Scientific) and Ham's F-12 Nutrient Mix (31756035; Thermo Fisher Scientific) supplemented with 10% FBS and 1% P/S. For the validation of gene editing, DNA was extracted from the cells using Quick-DNA™ Microprep Plus Kit (D4074; Zymo Research, CA, USA) according to the manufacturer's instructions. Then the gene sequencing was performed by Elim Biopharmaceuticals (CA, USA). Gene editing information for Be(2)-M17 cells is provided in the Supplementary Information file (Supplementary Fig. 17). Wildtype and Fis1 knock-out (KO) mouse embryonic fibroblasts (MEFs) were generously provided by Professor David C. Chan (California Institute of Technology)[2]. The cells were maintained in high-glucose DMEM supplemented with 10% FBS and 1% P/S. Fis1 KO was validated by immunoblotting with Fis1 antibody (10956-1-AP; Proteintech, IL, USA) (Supplementary Fig. 18). All cells were kept in a 95% air and 5% $CO_2$ atmosphere at 37 °C.

## Measurement of mitochondrial reactive oxygen species
HK-2 cells were plated in 96-well clear bottom black polystyrene microplates (3904; Corning, NY, USA) at $3.0 \times 10^4$ cells/well. The cells were incubated overnight in growth medium supplemented with 10% FBS, 1% P/S and 5 ng/ml hEGF. Then their media were removed and replaced with growth medium with or without 50 μM $H_2O_2$ (H1009; Sigma-Aldrich, MO, USA) with or without 250 nM SP11 or vehicle control (DMSO) and incubated for additional 24 h. Be(2)-M17 cells were plated in 96-well clear bottom black polystyrene microplates at a density of $1.0 \times 10^4$ cells/well. The cells were incubated overnight in growth medium supplemented with 10% FBS and 1% P/S. Then their media were removed, and replaced with serum-free MEM/F-12 with or without 50 μM $H_2O_2$ with or without 250 nM SP11 or vehicle control (DMSO) and incubated for an additional 24 h. Cells were washed with PBS and incubated in FluoroBrite™ DMEM (A18967-01; Thermo Fisher Scientific) containing 1 μM MitoSOX™ Red Mitochondrial Superoxide Indicators (M36008; Thermo Fisher Scientific) and 1 μg/mL Hoechst 33342 (H3570; Thermo Fisher Scientific) for 30 min at 37 °C and then washed with PBS and placed in FluoroBrite™ DMEM. Fluorescence was measured for MitoSOX™ (Ex 510 nm and Em 580 nm) and Hoechst 33342 (Ex 350 nm and Em 470 nm) using a fluorescent microplate reader SpectraMax M2 (Molecular Devices, CA, USA).

### Live-cell imaging for mitochondrial morphology

HK-2 cells were plated in 35-mm glass-bottom dish (P35GC-1.5-10-C; MatTek, MA, USA) at $2.0 \times 10^5$ cells/dish and were incubated overnight in growth medium. Cell media were then removed and replaced with glucose-free, serum-free, and galactose-supplemented (10 mM) DMEM/F12 medium (A2494301; Thermo Fisher Scientific) in the presence or absence of 100 μM $H_2O_2$ and in the presence or absence of SP11 or DMSO vehicle control, as above and incubated for 24 h. Cells were washed with PBS and incubated in FluoroBrite™ DMEM containing 100 nM MitoTracker Deep Red FM (M22426; Thermo Fisher Scientific) and 1 μg/mL Hoechst 33342 for 27 min at 37 °C. After PBS wash, cells were placed in FluoroBrite™ DMEM and Z-stacked images were randomly taken using a fluorescence microscope BX-X700 (Keyence, Osaka, Japan) outfitted with a stage top incubator (Tokai Hit, Shizuoka, Japan) using a 60X oil objective lens (MRD01605; Nikon, Tokyo, Japan). The Z-stacked images were further processed for the Z-projection and the haze reduction using Keyence BZ-Analyzer. The images from MitoTracker channel were binarized using ImageJ (version 1.54 f). Particles of each cell greater than 4 pixels were analyzed using ImageJ to exclude noise. The analysis was performed by a person blinded to the conditions.

### Isolation of mitochondria-enriched fraction

HK-2 cells were plated in 10 cm tissue culture plate (Genesee Scientific, CA, USA) at a density of $1.0 \times 10^6$ cells/plate and were incubated overnight in their growth medium. Cells were further incubated with serum-free growth medium for 24 h. Cell media were then removed and replaced with glucose-free, serum-free, galactose-supplemented (10 mM) DMEM/F12 medium with or without SP11 or vehicle control (DMSO) for 30 min. Then the cells were treated with 100 μM $H_2O_2$ in the presence of SP11 or vehicle control and incubated for additional 24 h. Cells were washed with PBS and harvested with a cell scraper. The collected cells were centrifuged briefly at $10,000 \times g$ for 1 min and the supernatant was discarded. The pellet was re-suspended with a 27-gauge 1/2-inch needle for lysis in mannitol-sucrose (MS) buffer, containing 210 mM mannitol, 70 mM sucrose, 5 mM MOPS (3-(N-morpholino) propane-sulfonic acid), 1 mM EDTA, and protease inhibitor cocktail (11697498001; Roche, Basel, Switzerland), followed by centrifugation at $800 \times g$ for 10 min at 4 °C to remove nuclei pellet. The supernatant was further centrifuged at $10,000 \times g$ for 20 min at 4 °C. The supernatant was collected for cytosolic fraction. The pellet was washed twice by gently re-suspending in MS buffer, and spinning down at $10,000 \times g$ for 5 min at 4 °C. This last supernatant was discarded, and the mitochondrial pellet was re-suspended in MS buffer containing 1% Triton X-100 (X100; Sigma-Aldrich), as before[38]. Note that in some experiments, there was no significant increased Drp1 association with the mitochondria 24 h after $H_2O_2$ treatment, relative to vehicle control. Thus, data presented in Fig. 7e and Supplementary Fig. 13, determining the effects of SP11 and C41S Fis1 on Drp1 association with the mitochondria, include only experiments in which $H_2O_2$-induced Drp1 translocation to the mitochondria was noted at that time point.

### Immunoblotting

Protein concentrations were determined using Pierce™ BCA Protein Assay Kit (23225; Thermo Fisher Scientific) following the manufacturer's instruction. Cell fractions from above were diluted in 4X Laemmli buffer containing DTT, heated at 70 °C for 10 min, loaded in 4–20% Mini-PROTEAN TGX™ Precast Protein Gels (Bio-Rad, CA, USA), and transferred to PVDF membrane (Bio-Rad). Membranes were incubated with indicated antibody and visualized by ECL (PI34095; Thermo Fisher Scientific). The images were acquired using Azure c600 imager (Azure Biosystems, CA, USA). The antibodies used in this study are: Anti-Drp1 (#8570; Cell Signaling Technology, MA, USA) at 1:500; Anti-VDAC (ab15895; Abcam, Cambridge, UK) at 1:1000; Anti-α/β Tubulin (#2148; Cell Signaling Technology) at 1:2000; Anti-Fis1 (10956-

1-AP, Proteintech) at 1:1000; β-actin (#3700, Cell Signaling Technology) at 1:5000; Anti-Rabbit IgG (NA934V; Cytiva) at 1:5000; Anti-Mouse IgG (NA931V; Cytiva) at 1:5000.

### Fis1 overexpression

For mitochondrial isolation and immunoblotting, Fis1 KO MEFs were plated at a density of $1.0 \times 10^6$ cells/plate in 10 cm tissue culture plate and were incubated overnight in their growth medium. The cells were then transfected with WT Fis1 (4000 ng/plate) or C41S Fis1 (8000 ng/plate) for 24 h. The cells were then treated with vehicle or $H_2O_2$ (50 μM) for 24 h in glucose-free and galactose-supplemented DMEM. Then Drp1 translocation to the mitochondria was analyzed using the same method as HK-2 cells. For mitochondrial imaging, Fis1 KO MEFs were plated at a density of $3.0 \times 10^5$ cells/plate in 35-mm glass bottom dish and were incubated overnight in their growth medium. The cells were then transfected with empty vector (280 ng/plate), WT Fis1 (140 ng/plate), or C41S Fis1 (280 ng/plate) for 24 h. The cells were then pre-treated with vehicle or SP11 (250 nM) for 30 min. Then vehicle or $H_2O_2$ (50 μM) was added, and the cells were incubated for additional 6 h. Then mitochondrial morphology in live cells was analyzed using the same method as HK-2 cells. We titrated the plasmid concentration to achieve equal protein expression in WT and C41S Fis1-transfected cells (Supplementary Fig. 18). Expression vectors were obtained from GenScript (NJ, USA). pcDNA3.1(+)-C-6His was used for empty vector. Sequences for the Fis1 inserts (cloned in HindIII/ApaI site of pcDNA3.1(+)-C-6His plasmid) are provided in the Supplementary Information file (Supplementary Fig. 19). Opti-MEM reduced serum medium (31985088; Thermo Fisher Scientific) and Lipofectamine 2000 reagent (11668019; Invitrogen) were used following manufacturer's instructions.

### Statistical Analysis

GraphPad Prism (version 10.4.1) was used to analyze all the data and generate graphs. T-test was used to determine the statistical difference between two groups and a one-way ANOVA with Dunnett's multiple comparisons or Tukey's multiple comparisons test was used for multiple groups.

## Data availability

The data that support this study are provided within the paper and associated supplementary files. Any raw and/or analyzed data that support the findings of this study are also available from the corresponding author upon request. Crystal structures of Fis1, Fis1 mutants and Fis1-SP11 co-crystal structures are deposited in the Protein Data Bank (PDB) under accession codes 9AVB, 9AVD, 9AVE, 9AYD, 9AVC, 9AYE. Source data are provided with this paper.

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

## Acknowledgements

This work was supported in part by DoD ALSRP Award W81XWH-22-1-0203 to D. M-R. Opinions, interpretations, conclusions and recommendations are those of the author and are not necessarily endorsed by the Assistant Secretary of Defense for Health Affairs or the Department of Defense. We thank SPARK at Stanford and SPARK advisors for their valuable support in discovery and development of inhibitors of pathological mitochondrial fission and Dr. Bruce Koch and Dr. Bereketeab Haileselassie, for their insightful suggestions on this project. We thank Dr. Daniel Fernandez and the Macromolecular Structure Knowledge Center at Stanford for providing material support for Fis1 crystallization. A. M. and S. Y. acknowledge support from JSPS KAKENHI (JP20H03230, JP22H04756 and JP23KJ1997) and JST SICORP Program (JPMJSC2203) in Japan. The numerical calculations were conducted in part using Cygnus and Pegasus at the Center for Computational Sciences, University of Tsukuba. G.H was supported by funding from the Stanford Medicine Children's Health Center for IBD and Celiac Disease. Use of the Stanford Synchrotron Radiation Lightsource, SLAC National Accelerator Laboratory, is supported by the U.S. Department of Energy, Office of Science, Office of Basic Energy Sciences under Contract No. DE-AC02-76SF00515. The SSRL Structural Molecular Biology Program is supported by the DOE Office of Biological and Environmental Research, and by the National Institutes of Health, National Institute of General Medical Sciences (P30GM133894). The contents of this publication are solely the responsibility of the authors and do not necessarily represent the official views of NIGMS or NIH. The Pilatus detector at beamline 4-2 at SSRL was funded under National Institutes of Health Grant S10OD021512.

## Author contributions

S.P., S.W. and D.M-R. conceived the study. S.P. designed the experiments. S.P., G.H., I.M., S.Y., A.M. and T.M., performed the experiments and analyzed the results. S.P., S.W. and D.M-R. wrote and revised the manuscript.

## Competing interests

S.P., S.W., and D.M-R. are listed as inventors on the provisional patent application on small molecules described in this study. The remaining authors declare no other competing interests.
