## [Transparent Peer Review file · Nature Communications]

A hidden cysteine in Fis1 targeted to prevent excessive mitochondrial fission and dysfunction under oxidative stress.

Corresponding Author: Professor Daria Mochy-Rosen

Version 0:

Reviewer comments:

Reviewer #1

(Remarks to the Author)

In the submitted manuscript entitled “A hidden cysteine in Fis1 targeted to prevent pathological mitochondrial fission and dysfunction”, the authors present data to show that residues on mitochondrial fission protein Fis1 (previously identified to activate Fis1 upon phosphorylation) allow for solvent exposure of the sole Cys residue (Cys41) in the molecule. Using recombinant wild type and phosphomimetic proteins purified to homogeneity, the authors show the phosphomimetics, but not WT, displaces or causes the disordering of helix 1. This is noteworthy as prior work has indicated that helix 1 might play an autoinhibitory role. The authors go on to show that the phosphomimetics allow for dimerization via Cys41.

Leveraging this knowledge, they screen a small molecule library that will covalently target this Cys to act as an inhibitor of activated Fis1. They identify one compound, SP11, a phenothiazine substituted chloroacetamide, that prevents dimerization in the T34D construct and also prevents H₂O₂ induced fragmentation of mitochondria consistent with the idea that SP11 prevents Fis1 recruitment of Drp1 for mitochondrial fission. However, this is the weakest part of the study as I note below. Since Fis1 has been reported to be activated only for stress-induced or pathological fission, the manuscript is conceptually exciting with the idea that a small molecule that targets only the activated form of Fis1. If correct, this may be a new example of targeting an activated, not resting, state of a protein target. Given that Fis1 is implicated in many different pathological scenarios, this is potentially quite exciting. However, the manuscript suffers from overinterpretation of the data in the title, abstract, and other sections. Many experimental details are insufficiently described to evaluate the rigor of the approach, despite the multiple complementary methods used in the biophysical aspects of the study. Finally, a major concern is that missing controls in the cellular data prevent one from determining whether reversal of peroxide induced mitochondrial fragmentation derives from Fis1 inhibition or off-target effects. Below I provide specific comments.

1. Instances of overinterpretation

Several claims in the abstract and throughout the manuscript are inferred and not well supported by the data:

1a. Title – treatment of HK cells with H₂O₂ is not considered pathological. It is arguable whether the Cys is hidden, see below.

1b. Abstract “In the presence of oxidative stress, the exposed Cys41 in activated Fis1 forms a disulfide bridge and the Fis1 covalent homodimers recruit Drp1 to mitochondria.”

The manuscript does not demonstrate this. The manuscript does demonstrate that treatment of recombinant protein with hydrogen peroxide favors dimer formation via a disulfide bridge (Extended Data Figure 4) and hydrogen peroxide treatment of cells causes mitochondrial fragmentation (7b) with a concomitant increase in mitochondrial Drp1 (7e), however, the mitochondrial fragmentation could also derive from an inhibition of mitochondrial fusion and no data (cellular or otherwise) shows Drp1 binding to Fis1 monomers or dimers, and a recent report indicates that Fis1 monomers bind to Drp1 (PMID: 37866629). The authors should either provide data that Fis1 dimers have higher affinity than Fis1 monomers for Drp1 or soften their language throughout to focus on what they do show.

1c. “confirmed the role of Fis1 homodimerization for Drp1 recruitment to mitochondria” See above comment.

SP11 does bind Fis1 and prevents homodimerization. This does not confirm the role of dimerization in Drp1 recruitment because no data was shown indicating that Fis1 homodimer was necessary for Drp1 binding. This is an inference and reasonable model to test in future studies, but the current data do not support this claim.

1d. “The idea of Cys41 on Fis1 being a sensor for oxidative stress” is indeed intriguing and worthy of investigation but is not

supported by data in the manuscript or in the literature. To demonstrate this, the authors would have to show that expression of a Cys-less Fis1 construct in Fis1 Δ cells did not respond to oxidative stress.

1e. Fig 1c shows a reduced helical propensity for residues 15-25 upon phosphorylation by MD simulations. The authors conducted thirty (!) 1.2 μ s simulations for this study, an impressive number but in the text they combine these runs into a single 36 μ s run, which is not valid and so the. The claim that phosphorylation reduces helical propensity is correct, however to state that their results indicate that these residues spend up to 799ns to 2318ns disordered is not supported by their data, the percentages need to taken from 1.2 μ s as it is not valid to concatenate the runs into a single 36 μ s run. The verbiage about the duration of disorder in lines 114-116 needs to be changed.

2. Missing controls

Figure 7 shows cell based data that H₂O₂ treatment causes mitochondrial fragmentation that is reversed upon treatment with SP11. To conclude that these derive from Fis1 inhibition, the authors need to repeat these experiments in Fis1^{-/-} cells, especially given that SP11 has a chloroacetamide warhead known to be promiscuous. Another missing control is also treatment of cells with SP11 in absence of H₂O₂.

3. WT Cys41: exposed or not?

The rationale that WT Cys41 is hidden and precluded from forming dimers is based on MD simulations (Fig 3b) and chemical modifications (Fig 3d) using a maleimide coumarin dye. The chemical modification data show that only T34D and Y38E are chemically modified whereas WT and Cys41Ser are not. These chemical modification data strongly supports the claim that the solvent accessibility of Cys41 increases upon phosphorylation (or, more correctly, phosphomimetic substitutions) but the interpretation that Cys41 is not solvent accessible is not justified as Fig 4b and Extended Figure 4 both show WT is capable of dimer formation by non-reducing SDS-PAGE. Yet the authors conclude "hence WT Fis1 could not form a covalent dimer Line 213-214". In Fig 4b, Y38E does show more dimer than WT, but the WT protein is able to form dimer to about the same extent as T34D by eye. Quantification of this gel is necessary to support the manuscript, and could help determine if the claims are correct. In Extended Data Figure 4, WT and phosphomimetics form dimers to about the same extent by eye. This gel should also be quantified to address this issue. No concentration-dependent data are presented, which would greatly strengthen their arguments given dimerization is at play. Time-dependent data could also address the issue of solvent accessibility. I suspect that Cys41 is solvent accessible – but to a lesser extent than the phosphomimetics – and the chemical modification data in Fig 3d arise from the relatively large size of the fluorescent maleimide probe. The authors should soften their language about the inaccessibility of Cys41 in WT Fis1 given their results.

4. Insufficient experimental details.

a. Figure 2a provides convincing crystallographic data that phosphomimetics T34D, T34E, and Y38E show no appreciable electron density for helix 1 compared to wild type, which supports the MD simulations suggesting the helix unfolding is possible upon phosphorylations.

To determine whether these results originated from crystalline artifacts the authors used size exclusion chromatography as detected by synchrotron small angle solution x-ray scattering data shown as a Kratky plot in Figure 2b. The uncertainties in the Porod volumes need to be reported. While the Kratky plot is often used to evaluate protein disorder, the authors should provide concentration dependent data for each construct with Guinier plots to show R_g , which would be expected to be different between WT and mutants and might be more informative than the Kratky plot shown. The Pair distribution function ($P(r)$ plots) would also be informative and could be included as supplemental. The authors should have already collected these data as it is routine for solution SAXS experiments and should provide these analyses.

b. Figure 5 shows that SP11, a small molecule discovered and reported in the manuscript, prevents the maleimide dye from binding T34D phosphomimetic. The data plateau at ~50% and is unexpected and troubling as one would expect that incubation with an equimolar concentration of SP11 should entirely modify the solvent accessible Cys in T34D. The authors ascribe this plateau to solubility or conformational heterogeneity. Neither of these explanations seem plausible. As SP11 reacts more should go into solution. Conformational heterogeneity might slow down the on-rate of SP11 but this is presumably an equilibrium measurement and therefore conformational heterogeneity cannot be at play. The origins of this should be determined and time dependent studies or protein concentration dependence could inform on this. What was the incubation time with the compound? Another possibility is that T34D is forming dimers preventing reactivity. Is SP11 labile or undergo oxidation? Have the authors tested this?

c. The x-ray structures appear to be of high quality but cannot be evaluated as indicated by the accompanying reports. The authors need to submit the PDB validation report.

5. Minor concerns:

a. The RMSF plot (1a) shows a helical periodicity for Y38-PO₄ suggesting that Helix 1 is still helical but simply might adopt 2 or more conformations. What do individual snapshots say about this? The data is consistent with 2 or more stable conformations being adopted. Do ϕ , ψ analyses support disruption of helix?

b. Fig1b – what structure is shown. What is starting structure for these trajectories? Were hydrogens used?

c. Figure 3 shows MD simulations and chemical modifications with CPM that show Cys41 is more solvent accessible upon PO4 of T34 or Y38.

A minor discrepancy exists between the MD simulations with T34 v Y38 and the chemical modification expts that show essentially no difference. A time course might resolve this.

What temp was the chemical modifications done at? If 25C then question is whether WT is modified at physiological temps.

d. Line 112 “36 μ s long MD simulations each” is potentially misleading.

e. Line 149 “Fis1 is activated under oxidative stress” is missing a logical step and would benefit from a bit more detail that included the oxidative stress conditions in the prior work (LPS) used to drive phosphorylation.

f. Consider replacing Figure 3b with a plot of the average SASA \pm sd that is reported in the text. This would be far easier to see. For this a p-value should be reported as well. If the authors prefer the current representation then the the figure legend should indicate what time interval is shown i.e. is the SASA data is calculated for each ns of the trajectory

g. Fig4a -consider placing residues 1-32 as dashed lines to indicate to the reader that no density was observed for these residues.

h. Line 891 consider replacing “shown” with “modeled”

i. Several of the PAGE images shown are referred to as “partial native PAGE” although the methods indicates that this was SDS-PAGE, which is non-native. This needs to be clarified. Do the authors mean non-reducing SDS-PAGE?

j. Line 218 “yeast Fis1 interaction with Caf4..... mediated by concave hydrophobic surfaces” is not quite right as the crystal structure of yeast Fis1 with Caf4 (2pqr.pdb) shows both concave and convex faces of Fis1 mediate the interaction. This paper should be cited (PubMed: 17998537).

k. Fgiure 5 legend should state the concentration of maleimide used. Line 245 could also state this as well.

l. Fig 5c – given the set up in Fig 5a and the manner in which the assay is presented in the text, Figure 5C may be better presented as a decrease in fluorescence as opposed to % inhibition.

m. Line 271 The argument of protein and SP11 concentration indicating a potent inhibitor could be better explained, especially given the covalent nature. It also contradicts what is stated in the Discussion that needs clarification as well.

n. Line348 “solution SAXS in solution “ should be changed to reflect what was done.

o. Fig 7e – from the figure legend and methods it is unclear at what stage the cells were treated with H2O2 and for how long.

p. The discussion meanders a bit and could be much tighter by focusing on the discoveries made and how it allowed the authors to devise a unique strategy to drug the “activated” form of a protein target. The evolutionary speculation with respect to avian Fis1 is quite interesting but seems out of place and perhaps better suited for a review.

q. Many experimental details are missing with respect to concetnratoins, incubation times, amounts of cells used, etc. Figure legends need updating with number of experiments and whether uncertainties are std dev or not. Gels need to be quantified and number of replicates indicated.

r. What cloning artifact does the protease cleavage leave. This should be stated and the length of the construct in the methods.

s. Fis1 has been reported to act via inhibition of the fusion machinery (PMID: 30842096), and since no data in the submitted manuscript eliminates this possibility, it should be incorporated into their model and discussion.

Blake Hill

Reviewer #2

(Remarks to the Author)

Review of NCOMMS-23-64230-T

A hidden cysteine in Fis1 targeted to prevent pathological mitochondrial fission and dysfunction by Pokhrel et al.

This is an interesting manuscript that characterizes novel conformational changes that permit redox sensitive crosslinking of Fis1, a receptor of Drp1 that promotes mitochondrial fission. This dimerization is observed in new crystal structures and the authors used MD to show that phosphorylation at specific sites can destabilize an N-terminal helix to allow for this crosslinking to occur. The studies are rigorous and the data novel. Overall, the most compelling part of the story is the identification of a novel Fis1 dimerization that could promote organelle fission in a redox-sensitive manner. The major limitation of the paper is the lack of direct evidence that this crosslink exists in cells. The in vitro work is solid and shows that the reactive Cys is engaged in dimerization. The use of a small molecule, SP11, inhibited this linkage through direct interactions with the Cys. Appropriate controls were used to show the selectivity of the moledule for this site, but these same controls were not pursued in cellular experiments that would definitively highlight the role of this linkage in promoting mitochondrial fission.

Otherwise, the flow of the paper is a little choppy and some suggestions are provided to help with any revisions.

1. The introduction is very terse. While I appreciate being direct and to the point, additional information could and should be provided to highlight the roles of Drp1 and Fis1 and expand on what is known about the Drp1-Fis1 interaction. This can highlight the gaps in knowledge since this paper seeks to modulate this interaction. I especially would recommend incorporating some discussion of previous structural studies with the human Fis1 to frame the impact of this study.
2. A representative movie comparing the fluctuations observed by MD in Figure 1 would be helpful. This would highlight the extent to which these changes are affecting the helical structure and movement.
3. The crystallography data are nice, but the absence of something (i.e. alpha1) is not proof of disorder. The SAXS analysis is nice, but NMR or some spectroscopy method would seem more appropriate. Others have had success with NMR of Fis1, and they did not see the movements in alpha1, but they could discern the flexibility in the N-terminal IDR.
4. Related to this previous point... the early mention highlighting that alpha1 is missing due to disorder seems a misleading when it is revealed that the construct formed a disulfide linked dimer in the lattice. This dimerization would seem incompatible with the formation of alpha1 (see Extended Fig 2). But the underlying cause for the loss of alpha1 could be stabilization of the crosslinked dimer. Do the densities presented in Fig 2 have the dimer pair density subtracted? I would highlight the dimerization earlier in the presentation of the structure and highlight the incompatibility with formation of the alpha1 helix when the protein forms this dimer. Currently, this gets lost and buried for me.
5. While the phosphomimetics offer compelling evidence for exposing Cys41, in vitro phosphorylation would provide validation that activation of the protein enhances Cys41 accessibility. Previous groups used this with Met to phosphorylate the protein in vitro at Y38, so this may be feasible.
6. The protein purification was done in the absence of any reducing agent, which may explain how the phosphomimetic proteins were able to form disulfide linked dimers in the crystal. But previous structural work with Fis1 were not oxidized, and this may explain why the destabilization of alpha 1 was observed. Can these dimers be induced after purifying reduced protein? How interchangeable are these interactions?
7. Related to the above point... Can these dimers be captured from endogenous Fis1 in the cell? Can they be captured to show that oxidation can lead to similar dimers without forcing them in vitro? The lack of cellular studies to identify the disulfide crosslink is a major limitation. The authors used magnetic beads to force His-tagged construct into close proximity. But could this cross-link be seen in cells? Organlle association could be driven by a number of independent factors. In addition to H2O2, did the authors used the phosphomimetic proteins to enrich for crosslinks that could be captured? Without this evidence, these data could represent an in vitro phenomenon.
8. It is unclear to me where the SP22 analogue came from or how it serves as a control for the SP11 experiments. It appears to behave similarly, though maybe with less potency.
9. Do the SAXS experiments have the resolution to reliably identify a conformational change in a single alpha helix in this small protein? The envelope seems similar, so I am not sure where the difference is predicted. Also, P2 symmetry was implemented. Were densities examined without symmetry applied to see what the resulting maps looked like? It may be beneficial to test this.
10. A major limitation in the field is the inability to reproducibly observe Drp1-Fis1 interactions with isolated proteins. Did the authors attempt to examine the impact of Fis1 dimerization on Drp1 activity and/or structure? This would help define the impact of this change on mitochondrial fission.
11. Do we know whether the SP11 interacts with H2O2 directly? Addition of both in the same media may directly lead to less oxidation if there is an interaction with H2O2. It would be nice to identify a control that would be insensitive to SP11 and show that it is still leading to ROS stress when treated with both H2O2 and SP11. Any evidence to show that there is no interaction would suffice tbh.

Minor comments:

12. The mention of SP11 in the Abstract confusing, since this has not been defined. I would note that this is a small molecule designed as an inhibitor for Drp1 in this section.
13. The dimerization that was compared in the paper (ref 17, line 182) was looking for a different, inhibitory dimerization. This was identified in the Yeast Fis1 protein several years ago (Lees et al, JMB, 2012). Some comparison with this model would help define the unique attributes of this dimerization interface through Cys41 and highlight why this could be productive, rather than inhibitory. In fact, this comparison would highlight regions that would be available for interactions with Drp1 directly.
14. The legend for Extended Figure 6 explains things for SP11 instead of SP22, which the compound used in these experiments I think.

(Remarks to the Author)

In this study, the authors investigate Fis1 activation in response to stress through structural changes alterations that mediate Drp1 binding on the mitochondria. This regulatory mechanism while poorly defined is important because Fis1-Drp1 interaction mediates pathological mitochondrial fission. The N-terminal region of Fis1 is dynamic and the author speculate that perturbations such as phosphorylation at key tyrosine on threonine residues on the 2-helix cause unfolding to expose the single cysteine residue C41. Oxidized C41, presumably through sensing of oxidative stress, mediates Fis1 covalent dimerization and activation. The authors use a combination of MD simulations, site-directed mutagenesis and structural biology to support this intriguing regulation.

Next, the authors screen for cysteine-reactive fragments to bind and block Fis1 covalent dimerization activation and identify a lead compound SP11, a phenothiazine-chloroacetamide, that bound C41 on activated Fis1. The authors also test an acrylamide counterpart (SP22) that show comparable effects. Site of binding for SP11 was confirmed by co-crystal structures and LC-MS/MS. This manuscript displays an impressive array of diverse methods to tackle this challenging problem. The identification of covalent inhibitors of Fis1-Drp1 protein-protein interaction (PPI) via disruption of a disulfide is important for guiding PPI inhibitor development. Fis1 inhibitors could also have therapeutic applications.

While the studies are promising, I have a major issue with the lack of selectivity studies in cellular studies using SP11 and the claims made with regards to Fis1 function using this compound. While I appreciate the authors introducing a second cysteine site to counter screen for non-specific binders, these studies are still only testing covalent binding of compounds against a single protein. Chemical proteomic methods are available to evaluate selectivity of cysteine-binding ligands such as SP11 and authors should use these methods to provide evidence of reasonable selectivity in SP11-treated cells. For example, alkyne-tagged iodoacetamide is an effective probe for evaluating ligand binding to cysteines proteome-wide. Additionally, I see no evidence for target engagement of Fis1 by SP11 in cellular studies. These studies are important for supporting the claims that the mitochondrial phenotypes observed in cellular studies are due to disruption of Fis1-Drp1. I suggest including a cysteine-reactive but Fis1-inactive compound as a negative control to further support Fis1 on-target effects.

Additional considerations:

- Figure 4b – to further support covalent dimers in solution, can authors add BME to disrupt and evaluate by native gel?
- Claim of SP11 having drug-like characteristics in the discussion section is a bit overstated given there are no data supporting this statement.
- Authors claim SP11 has excellent potency in the discussion. This claim should be supported by a measurement of potency for covalent inhibitors. For example, a measure of k_{inact}/K_I would be appropriate.
- Figure 6A: MS/MS spectra shows many peaks that are not annotated. Can the authors explain? How confident are authors in the identification? Other metrics used to further support quality of the modified peptide?

Version 1:

Reviewer comments:

Reviewer #1

(Remarks to the Author)

The revised manuscript entitled “A hidden cysteine in Fis1 targeted to prevent excessive mitochondrial fission and dysfunction under oxidative stress” is significantly improved and many of the results interpreted appropriately and new data has been added to address the majority of referees’ concerns, although a few lingering issues exist:

1. Specificity of SP11.

Previously, the authors had reported that treatment of HK-2 cells with H₂O₂ increased mitochondrial fragmentation, mitochondrial reactive oxygen species (mtROS, as measured using the MitoSOX probe), and Drp1 recruitment to mitochondria. The authors went on to show that pre-treatment of the SP11 compound prevented the H₂O₂-induced effects. These data were presented in the originally submitted manuscript, but did not address whether SP11 was acting by targeting Fis1. Given the reactive nature of SP11, this is an important issue raised by referees. To address this, the authors used a Cys41Ser knock-in cell line and new data (Fig 7f) show no increase in mtROS upon treatment with H₂O₂, which supports the claim that mtROS increase in WT is generated by Fis1 oxidation at Cys41. Treatment with the SP11 compound under these conditions has no effect suggesting that off-target effects of SP11 are not responsible for the decrease in mtROS generation. However, mitochondrial morphology and Drp1 recruitment were not examined in the new cell line, and would strongly support their claims. The rebuttal mentions that Drp1 recruitment was examined in this new cell line...perhaps the data were mistakenly not included?

2. Helix 1 unfolding

The model the authors propose is that phosphorylation of either Thr34 or Tyr38 unfolds helix 1, exposing Cys41, causing disulfide-mediated dimerization of Fis1 (Cys41 is the sole Cys in Fis1), which allows for enhanced Drp1 recruitment. The data largely support the model except for helix 1 unfolding and Fis1 dimer, not monomer, binding Drp1 (point #3 below). The evidence that helix 1 is unfolded is not compelling, and it is not necessary. The unfolding interpretation is based on 3 observations: (1) MD simulations (2) lack of density in x-ray crystallography, and (3) SAXS data. The MD does show local

unfolding is more possible in the pT34 and pY38 constructs compared to wild type, however, the phi/psi analyses and the DSSP analyses suggest a modest unfolding is possible, and don't support the claims that helix 1 unfolds. As Referee 2 pointed out, absence of density in x-ray crystallography is consistent with unfolding but also consistent with other types of disorder less severe. SAXS data has the possibility to resolve these issues and the P(r) plots are nearly superimposable with a longer tail for WT and T34D than Y38E (reaches zero at ~60Å for WT and ~56Å for Y38E). This is opposite one would expect for helix 1 unfolding in the phosphomimetics compared to wild type. Additionally, the AUC under the curve from the Kratky plot is consistent with Y38E being more compact as it is the most symmetric curve with the lowest AUC value under 4. The Porod volume is larger for Y38E and this likely is the basis for their interpretation. Perhaps plotting their existing data as $q^3I(q)$ vs q^3 or $q^4I(q)$ vs q^4 could address this, but to me these data are indistinguishable. My suggestion is that the authors consider replacing the occurrences of helix 1 unfolding with a conformational change involving helix 1 (that exposes Cys41). This is exactly what Extended Data Figure 2 shows, and is the most consistent interpretation of their data. If the authors insist on interpreting their data as unfolded then CD, or better, NMR, should be provided to support the claims. Alternatively, limited proteolysis could be used to probe the lability of the N-terminal region of WT vs phosphomimetics.

3. Is Fis1 dimerization necessary for Drp1 interaction

The manuscript would be improved mechanistically if the authors could determine whether Fis1 dimer directly interacts with Drp1. This group previously published in Nature Communications using recombinant Drp1 and so they can produce both proteins and test for binding. I respect that they want to reserve this for future work, but it would improve their model. I note that their statement to Referee #3 that prior work has not established a Fis1-Drp1 direct interaction neglects our work (see PMID:37866629), which I pointed out in my earlier review. The authors are correct in that other groups (Jason Mears and Jodi Nunnari, personal communication) have not observed a direct interaction, but I think this is because it is autoinhibited by the N-terminal region as we stated in our JBC papers on this subject (only one of which was referenced, PMID: 37866629 and 37777154 are not). Given the authors model in which the N-terminal region prevents exposure of Cys41, it seems appropriate that this prior work incorporated in their discussion.

4. Discussion

The revised discussion deleted the first paragraph that placed their work in context of earlier work (including ours. Ref 18) showing an important role for the Fis1 N-terminal region. It is unclear why this paragraph was deleted. As noted above (#3), the authors should also incorporate PMID: 37866629 and 37777154 in their discussion given the important role of the N-terminus to their model.

Minor points-

5. Using the terminology partial native SDS-PAGE is confusing and inaccurate. SDS is a denaturant and it implies something different than what was done, despite Hwang et al usage. As described this is non-reducing SDS-PAGE in which the samples are not boiled, a common practice in reducing and non-reducing SDS-PAGE. Hwang et al is the only reference I could find that has this terminology and this reference is from the same lab, so I do not see this as a generally accepted terminology.

6. Line 1010 – porod should be capitalized

7. Line 572 – percent is misspelled. The document should be checked for other misspellings.

8. CPM Assay – the descriptions of concentrations in the CPM assay are confusing. What is meant by test molecule? Stating whether this is CPM itself or the protein being tested would clarify.

9. Reporting an IC50 is unclear. Is this from equilibrium measurements? i.e. waiting ~62 hours? If not, then it should be reported as an "apparent IC50".

Reviewer #2

(Remarks to the Author)

The authors did a thorough job in addressing the reviewers comments throughout, and I appreciate the effort that went into the additional data that is now presented.

The changes in the text are thoughtful and largely address the reviewers' concerns.

There are inherent limitations with this system, but this work represents an important advance for the field. I am in favor of acceptance.

Reviewer #3

(Remarks to the Author)

I appreciate the authors inclusion of new data using C41S knock-in M17 cells to support target engagement with SP11 but the major critique regarding selectivity across the proteome from cellular studies is still not addressed. I believe this key

question must be addressed since many of the claims are based on the ability of SP11 to selectively block covalent dimerization and activation for investigating Fis1 function. Unfortunately, I am not able to support publication till this major critique has been properly resolved.

Version 2:

Reviewer comments:

Reviewer #1

(Remarks to the Author)

In the revised manuscript, the authors have addressed my earlier concerns with their changes. I do agree with Rev #3 that off target effects are a concern and the new Fis KO MEF data are a welcome and important addition to the study. In these new data, SP11 appears to have an effect in the absence of Fis1 with changes in the morphology scores upon addition to Empty Vector after treatment with H₂O₂ (7g, from 18 to 8% fragmentation) and increases from 15 to 34% of fused mitochondria upon SP11 treatment in C41S cells. These effects indicate some off target effects do occur. I think they are engaging the target Fis1 based on the reduction of their other data including the Fis1-dependent fragmentation shown in this figure 7g (reduction from 50 to 155). However, I think they should mention the off target effects and soften their language throughout the manuscript. Overall, a very impressive study that advances the field.

Reviewer #3

(Remarks to the Author)

My question regarding selectivity in proteomes from cellular studies is still not fully addressed but the new Fis1 KO MEF data with transfected C41S Fis1 mutant provides supporting evidence for on-target activity.

Reviewer #4

(Remarks to the Author)

The manuscript by Pokhrel et al. describes how a signal triggered by oxidative stress is amplified through Fis1 protein dimerization, leading to the modulation of mitochondrial fission. The work combines a wide range of experimental techniques with atomistic molecular dynamics (MD) simulations.

I find the results broadly consistent, with the MD simulations supporting the other techniques. Technically, the simulations are sound, using well-established algorithms and simulation parameters suitable for addressing the proposed hypothesis. Overall, I found no major issues with them. The description in the Methods section allows for replication of the simulation results. One minor issue is that it is not specified how different trajectory replicas were generated (although this information is provided in the editorial MD checklist).

**Reviewer #1** (Remarks to the Author):

In the submitted manuscript entitled “A hidden cysteine in Fis1 targeted to prevent pathological
mitochondrial fission and dysfunction”, the authors present data to show that residues on mitochondrial
fission protein Fis1 (previously identified to activate Fis1 upon phosphorylation) allow for solvent exposure
of the sole Cys residue (Cys41) in the molecule. Using recombinant wild type and phosphomimetic proteins
purified to homogeneity, the authors show the phosphomimetics, but not WT, displaces or causes the
disordering of helix 1. This is noteworthy as prior work has indicated that helix 1 might play an
autoinhibitory role. The authors go onto show that the phosphomimetics allow for dimerization via Cys41.

Leveraging this knowledge, they screen a small molecule library that will covalently target this Cys to act
as an inhibitor of activated Fis1. They identify one compound, SP11, a phenothiazine substituted
chloroacetamide, that prevents dimerization in the T34D construct and also prevents H₂O₂ induced
fragmentation of mitochondria consistent with the idea that SP11 prevents Fis1 recruitment of Drp1 for
mitochondrial fission. However, this is the weakest part of the study as I note below. Since Fis1 has been
reported to be activated only for stress-induced or pathological fission, the manuscript is conceptually
exciting with the idea that a small molecule that targets only the activated form of Fis1. If correct, this may
be a new example of targeting an activated, not resting, state of a protein target. Given that Fis1 is implicated
in many different pathological scenarios, this is potentially quite exciting. However, the manuscript suffers
from overinterpretation of the data in the title, abstract, and other sections. Many experimental details are
insufficiently described to evaluate the rigor of the approach, despite the multiple complementary methods
used in the biophysical aspects of the study. Finally, a major concern is that missing controls in the cellular
data prevent one from determining whether reversal of peroxide induced mitochondrial fragmentation
derives from Fis1 inhibition or off-target effects. Below I provide specific comments.

1. Instances of overinterpretation: Several claims in the abstract and throughout the manuscript are inferred
and not well supported by the data:

1a. Title – treatment of HK cells with H₂O₂ is not considered pathological.

*Ans: We agree and have changed the title to “A hidden cysteine in Fis1 targeted to prevent excessive
mitochondrial fission and dysfunction under oxidative stress.”*

It is arguable whether the Cys is hidden, see below.

*We have presented several lines of evidence to suggest that the Cys41 in Fis1 under basal condition has a
lower solvent exposure and ability to form covalent homodimer compared to the phosphorylation mimic
mutants. These include data from: (A) molecular dynamics simulations where Cys41 of phosphorylation
mimic mutants have higher solvent accessible surface area compared to the WT, (B) CPM fluorescence
assay where phosphorylation mimic mutants produced about 12 fold higher CPM fluorescence compared
to the WT, (C) X-ray crystallography where phosphorylation mimic mutants crystallized as covalent dimer
whereas WT didn't, and (D) partial native page where immobilized phosphorylation mimic mutants had
high propensity to form dimer compared to the WT.*

*With further support from the new evidence from Cys41Ser knock in homozygous M17 cells, we hope that
reviewer agrees that the Cys41 residue is critical and is hidden/unavailable in Fis1 under non-oxidative,
basal state. These data are presented in the subsequent responses.*

1b. Abstract “In the presence of oxidative stress, the exposed Cys41 in activated Fis1 forms a disulfide
bridge and the Fis1 covalent homodimers recruit Drp1 to mitochondria.” The manuscript does not
demonstrate this. The manuscript does demonstrate that treatment of recombinant protein with hydrogen
peroxide favors dimer formation via a disulfide bridge (Extended Data Figure 4) and hydrogen peroxide
treatment of cells causes mitochondrial fragmentation (7b) with a concomitant increase in mitochondrial
Drp1 (7e), however, the mitochondrial fragmentation could also derive from an inhibition of mitochondrial
fusion and no data (cellular or otherwise) shows Drp1 binding to Fis1 monomers or dimers, and a recent
report indicates that Fis1 monomers bind to Drp1¹ (PMID: 37866629). The authors should either provide

data that Fis1 dimers have higher affinity than Fis1 monomers for Drp1 or soften their language throughout
to focus on what they do show.

Ans: We thank the reviewer for pointing this out, added new data using Cys41Ser knock in mutant (see
below), and have modified the abstract accordingly:

“In the presence of oxidative stress, the exposed Cys41 in activated Fis1 forms a disulfide bridge and the
Fis1 covalent homodimers cause increased mitochondrial fission through increased Drp1 recruitment to
mitochondria.”

The following data support the modified statement. SP11 treatment in HK-2 cells that prevents Fis1
covalent dimerization through Cys41 results in reduced Drp1 localization to mitochondria (Fig. 7e) and
Cys41Ser knock in homozygous M17 cells have reduced sensitivity to oxidative stress as seen in the
MitoSOX assay. Therefore, the SP11 effect is likely by binding Cys41 in WT Fis1 and not through other
mechanisms. The new data are in Fig. 7f.

As pointed out by the reviewer, a recent study demonstrated that Fis1 monomers binds to Drp1 in vitro¹.
We predict that Fis1 covalent dimers bind Drp1 with a higher affinity compared to the monomers, although
this was not determined in our study. However, we show that Fis1 that lacks Cys41 for dimerization is
ineffective in recruiting Drp1 in cells, supporting (but not proving) that the dimer form of Fis1 is likely the
active form of Fis1 that can recruit Drp1 to the mitochondria. Further biophysical and structural
characterization of Drp1 binding to Fis1 monomer and covalent dimers will be investigated in future
studies.

1c. “ confirmed the role of Fis1 homodimerization for Drp1 recruitment to mitochondria” See above
comment. SP11 does bind Fis1 and prevents homodimerization. This does not confirm the role of
dimerization in Drp1 recruitment because no data was shown indicating that Fis1 homodimer was necessary
for Drp1 binding. This is an inference and reasonable model to test in future studies, but the current data do
not support this claim.

Ans: The reviewer is correct and thus added new data using Cys41Ser knock in homozygous M17 cells,
showing that these cells have reduced sensitivity to oxidative stress as seen in the MitoSOX assay. We also
show that Drp1 recruitment to mitochondria is reduced under oxidative stress induced by H₂O₂ in HK-2
cells treated with SP11 that binds to Cys41, the only cysteine residue in Fis1. These data suggest that Cys41-
mediated Fis1 covalent homodimer is necessary for Drp1 recruitment to mitochondria.

We have modified the abstract to reflect these data:

“Our discovery of a small molecule, SP11, that binds only to activated Fis1 by engaging Cys41, and data
from genetically engineered cell lines lacking Cys41, strongly suggest a role of Fis1 homodimerization in
Drp1 recruitment to mitochondria and excessive mitochondrial fission.

The new data are included in the manuscript as Fig. 7f.

1d. “The idea of Cys41 on Fis1 being a sensor for oxidative stress” is indeed intriguing and worthy of
 investigation but is not supported by data in the manuscript or in the literature. To demonstrate this, the
 authors would have to show that expression of a Cys-less Fis1 construct in Fis1Δ cells did not respond to
 oxidative stress.

Ans: Indeed, the data discussed above (new Fig. 7f) show that Cys-less Fis1 construct-containing cells do
 not respond to oxidative stress; mitochondrial ROS levels are unaffected.

1e. Fig 1c shows a reduced helical propensity for residues 15-25 upon phosphorylation by MD simulations.

The authors conducted thirty (!) 1.2 μs simulations for this study, an impressive number but in the text they
 combine these runs into a single 36μs run, which is not valid and so the claim that phosphorylation reduces
 helical propensity is correct, however to state that their results indicate that these residues spend up to 799ns
 to 2318ns disordered is not supported by their data, the percentages need to taken from 1.2 μs as it is not
 valid to concatenate the runs into a single 36 μs run. The verbiage about the duration of disorder in lines
 114-116 needs to be changed.

Ans: We have modified lines 114-116 (new lines 119-125) as follows, to explain the basis for these claims:

“We calculated the average propensity of residues in α1 helix (aa 15-25) to form α-helical structure using
 the DSSP algorithm in 30 independent 1.2 μs-MD simulations for each of WT, pThr34, and pTyr38.
 Compared to unphosphorylated Fis1 (WT), the propensity of amino acid residues 15-25 to form α-helical
 structure in pThr34 and pTyr38 were lower by 0.19-2.22% and 1.91-6.44%, respectively (Fig. 1c, Extended
 Data Fig. 1, Extended Data Movie 1). We also performed the φ and ψ angle analysis in these simulations.
 We found that there were differences in standard deviations, but the differences in the average φ and ψ
 values were smaller (Extended Data Table 3).”

The above data are included in the manuscript as Extended Data Fig. 1.

2. Missing controls

Figure 7 shows cell based data that H₂O₂ treatment causes mitochondrial fragmentation that is reversed upon treatment with SP11. To conclude that these derive from Fis1 inhibition, the authors need to repeat these experiments in Fis1^{-/-} cells, especially given that SP11 has a chloroacetamide warhead known to be promiscuous. Another missing control is also treatment of cells with SP11 in absence of H₂O₂.

Ans: As indicated above, we have generated Cys41Ser knock in homozygous M17 cells and found that these cells are less sensitive to H₂O₂-induced oxidative stress and that treatment with SP11 has no effect on MitoSOX signal, unlike cells expressing WT Fis1 (Fig. 7f).

We also show that SP11 has no effect in non-stimulated cells in MitoSOX assay (See below and included in the manuscript as Extended Data Fig. 8).

3. WT Cys41: exposed or not?

The rationale that WT Cys41 is hidden and precluded from forming dimers is based on MD simulations (Fig 3b) and chemical modifications (Fig 3d) using a maleimide coumarin dye. The chemical modification data show that only T34D and Y38E are chemically modified whereas WT and Cys41Ser are not. These chemical modification data strongly supports the claim that the solvent accessibility of Cys41 increases upon phosphorylation (or, more correctly, phosphomimetic substitutions) but the interpretation that Cys41

is not solvent accessible is not justified as Fig 4b and Extended Figure 4 both show WT is capable of dimer
formation by non-reducing SDS-PAGE. Yet the authors conclude “hence WT Fis1 could not form a covalent
dimer Line 213-214”.

Ans: We thank the reviewer for pointing this out. The differences in the solvent exposure of Cys41 in the
WT and phosphorylation mimic mutants are apparent in the molecular dynamics simulations and chemical
modification experiments. However, as rightly pointed out, we see some covalent dimerization propensity
of WT Fis1 when we analyze the recombinant proteins on a partial native SDS-PAGE. However, no
dimerization of WT is seen when we immobilize the same amounts of WT and phosphorylation-mimic
mutant proteins and incubate in a buffer with 10 mM 2-ME for an hour (to break preexisting covalent
dimers) followed by a wash out of 2-ME. Incubation at room temperature that induces a time-dependent
dimerization for the mutants, but not for the WT Fis1. This experiment confirms that WT Fis1 has a much
lower propensity to form covalent dimer compared to the phosphorylation mimic mutants. The new data
are provided below and Fig. 4b has been replaced with these new data.

We have also added a new sentence after “hence WT Fis1 could not form a covalent dimer Line 213-214”
statement.

“We rationalized that the α 1-helix and N-terminus structure is intact in WT Fis1, making Cys41 unavailable
for disulfide bridge formation and hence WT Fis1 could not form a covalent dimer (Extended Data Fig. 3).
However, due to the inherent dynamic nature of N-terminus of Fis1 as seen in MD simulations and CPM
fluorescence assay (Fig. 1a, 3d), WT Fis1 showed some propensity to form covalent dimers (Extended Fig.
5). We observed that the concave hydrophobic surface that is known to mediate protein-protein interactions
in TPR motif-containing proteins is between the interface of the dimer in both our current and the previously
reported structure (Fig. 4d).”

In Fig 4b, Y38E does show more dimer than WT, but the WT protein is able to form dimer to about the
same extent as T34D by eye. Quantification of this gel is necessary to support the manuscript, and could
help determine if the claims are correct. In Extended Data Figure 4, WT and phosphomimetics form dimers
to about the same extent by eye. This gel should also be quantified to address this issue. No concentration-
dependent data are presented, which would greatly strengthen their arguments given dimerization is at play.
Time-dependent data could also address the issue of solvent accessibility. I suspect that Cys41 is solvent
accessible – but to a lesser extent than the phosphomimetics – and the chemical modification data in Fig 3d
arise from the relatively large size of the fluorescent maleimide probe. The authors should soften their
language about the inaccessibility of Cys41 in WT Fis1 given their results.

Ans: The experiment suggested by the reviewer was included and the result and quantification are provided
above and the new Fig. 4b.

Regarding Extended Data Fig. 5, we did observe some non-cysteine mediated covalent cross linking (as in
the case of Cys41Ser Fis1 construct without any Cysteine) and therefore we cannot directly compare the

extent of Cys41-mediated covalent dimerization across different Fis1 constructs in this experimental
 condition. This experiment was performed in the presence of high concentration (20 mM) of hydrogen
 peroxide which might have also caused artifact. Following reviewer's suggestion, we have now added a
 more extensive time-dependent data (and quantitation) to show that phosphorylation mimic mutants have
 much higher propensity to form covalent dimers compared to WT. We have now replaced Extended Data
 Fig. 5 with this new figure below.

4. Insufficient experimental details.

a. Figure 2a provides convincing crystallographic data that phosphomimetics T34D, T34E, and Y38E show
 no appreciable electron density for helix 1 compared to wild type, which supports the MD simulations
 suggesting the helix unfolding is possible upon phosphorylations. To determine whether these results
 originated from crystalline artifacts the authors used size exclusion chromatography as detected by
 synchrotron small angle solution x-ray scattering data shown as a Kratky plot in Figure 2b. The uncertainties
 in the Porod volumes need to be reported. While the Kratky plot is often used to evaluate protein disorder,
 the should provide concentration dependent data for each construct with Guinier plots to show R_g , which
 would be expected to be different between WT and mutants and might be more informative than the Kratky
 plot shown. The Pair distribution function ($P(r)$ plots) would also be informative and could be included as
 supplemental. The authors should have already collected these data as it is routine for solution SAXS
 experiments and should provide these analyses.

Ans: The Porod volumes reported in Table 1 were computed using the program Primus (GNOM applet).
 However, the algorithmic details employed by the program to estimate the Porod Volume remain
 undisclosed to the public. As the reviewer pointed out, the Porod Volume is sensitive to the sample's
 globularity (flexibility) and shape, but not invariably sensitive to lower angle information. The Porod
 Volume is frequently utilized for M_w estimation, and it is a widely accepted as a rule of thumb in the
 BioSAXS community that the estimation error would be approximately +/- 15%. Consequently, it is
 challenging to evaluate the uncertainties of the Porod volume as it is a number dependent on the sample's
 globularity and shape.

Subsequently, to assess the data sets, we computed the Porod Volumes with different D_{max} values (+/- 5Å)
 using Primus (data shown below and added as Supplementary Fig. 1a in the manuscript). The q-range used
 for this analysis was fixed for comparisons ($0.016 < q < 0.45$), whereas the q-ranges of each sample in
 Figure 2 were optimized. The Porod volumes exhibited no sensitivity to the D_{max} values, and the coefficients

of variance in all three samples were less than 0.8% (S.D. is less than 200\AA^3). This may suggest that the
 data sets do not incorporate any structural factors from inter-molecular interactions, but that minor
 alterations in the Porod Volume may correspond to different folding states. Since we are unaware of how
 the Porod Volume was computed by Primus, we are hesitant to discuss the uncertainties of the Porod
 volumes in the manuscript.

	WT	T34D	Y38E
$D_{\max} - 5\text{\AA}$	24310	25452	28421
D_{\max}	24508	25794	28630
$D_{\max} + 5\text{\AA}$	24531	25795	28714
Average Volume	24450	25680	28588
SD	121	198	151
% of SD	0.50	0.77	0.53

Even though we have tried our best to answer the reviewer's questions, we feel that this analysis doesn't
 sufficiently address the reviewer's concerns largely due to the lack of availability of the algorithmic details
 of the program. To circumvent this, we have now added a new analysis, where we have calculated the area
 under the Kratky plot (unscaled Porod invariant) for each construct which is inversely proportional to the
 sample volume. We observe that this analysis is consistent with the calculations reported for the porod
 volume; we see that the area under the Kratky plot for the phosphorylation mimic mutants T34D and Y38E
 is lower compared to WT Fis1, suggesting larger porod volumes (data shown below). We have integrated
 only the area of the plot up to the $qR_g = 4$ where the signal is less noisy. We have now replaced Fig. 2b with
 the figure below.

We have incorporated the SEC-SAXS plot (image number vs R_g and $I(0)$) in Supplementary Fig. 1b and
 below. We obtained reasonably flat R_g plots around the main peak where the concentrations are different
 as shown by A280 in all three samples, indicating no significant concentration-dependence. It should be
 noted that the deviation from the plateau on both sides of the peak is attributable to the challenges of scaling
 individual images and subsequent background subtraction at very low concentrations.

WT Fis1

Thr34Asp Fis1

Tyr38Glu Fis1

 The P(r) plots have also been included in Supplementary Fig. 1c and is shown below. We greatly appreciate
 the reviewer's comment, as the addition of these figures enhances the reliability of the Kratky plot
 interpretations.

WT Fis1

Thr34Asp Fis1

Tyr38Glu Fis1

 b. Figure 5 shows that SP11, a small molecule discovered and reported in the manuscript, prevents the
 maleimide dye from binding T34D phosphomimetic. The data plateaus ~50% and is unexpected and
 troubling as one would expect that incubation with an equimolar concentration of SP11 should entirely
 modify the solvent accessible Cys in T34D. The authors ascribe this plateau to solubility or conformational
 heterogeneity. Neither of these explanations seem plausible. As SP11 reacts more should go into solution.
 Conformational heterogeneity might slow down the on-rate of SP11 but this is presumably an equilibrium
 measurement and therefore conformational heterogeneity cannot be at play. The origins of this should be
 determined and time dependent studies or protein concentration dependence could inform on this. What
 was the incubation time with the compound? Another possibility is that T34D is forming dimers preventing
 reactivity. Is SP11 labile or undergo oxidation? Have the authors tested this?
 Ans: As mentioned by the reviewer, the inhibition data plateaus at about 50% (at 30 min incubation time).
 This was unexpected. But this can be explained by the conformational heterogeneity. To reiterate, the way
 this assay works is that the protein is first incubated with SP11 and this binding is heavily dependent upon
 the conformation of the pocket. Whereas in the second step, the maleimide-based probe is added to the
 mixture and maleimide binds to the unoccupied Cys41 and produces fluorescence. This maleimide-based
 probe binding doesn't depend on the conformation as long as the Cys41 is exposed. Cys41 exposure isn't
 sufficient to bind SP11 and this is further confirmed by inability of SP11 to binding the Tyr38Glu Fis1 with
 exposed Cys41. Based on MD simulations data, it can be presumed that Thr34Asp Fis1 shows a distribution
 of conformations with exposed Cys41 in solution that can bind maleimide-based dye but not all of these
 conformations can bind SP11. Therefore, the maximal inhibition is limited by the fraction of Thr34Asp Fis1
 in favorable conformation to bind SP11 which may not be 100%. This observation was similar in both SP11
 and SP22. To confirm this, as suggested by the reviewer, we performed the time course experiment, and we
 observed that the percent inhibition approaches 100% with the longer incubation time. This strongly

suggests the conformational heterogeneity and dynamics of Thr34Asp Fis1. The data are provided below
and included in the manuscript as Supplementary Fig. 2.

We do not think that Thr34Asp dimer formation contributed to this reduction in maximal inhibition as
neither SP11 nor maleimide based dye can bind to Thr34Asp Fis1 covalent dimer and therefore the signal
is unaffected.

We have measured the chemical stability of SP11 in PBS pH 7.4 using LC-MS/MS. We didn't observe any
modifications on SP11 for over 7 days. The data is shown below.

c. The x-ray structures appear to be of high quality but cannot be evaluated as indicated by the
accompanying reports. The authors need to submit the PDB validation report.

Ans: X-ray structures have been deposited to PDB and the PDB validation reports for the structures are
provided.

5. Minor concerns:

a. The RMSF plot (1a) shows a helical periodicity for Y38-PO4 suggesting that Helix 1 is still helical but
simply might adopt 2 or more conformations. What do individual snapshots say about this? The data is
consistent with 2 or more stable conformations being adopted. Do phi,psi analyses support disruption of
helix?

Ans: The reviewer is correct; as reported in Extended Data Figure 1, $\alpha 1$ helix mainly formed α helix in the
MD simulations for pT34 and pY38. However, the $\alpha 1$ helix was clearly disrupted compared to the WT-
Fis1, and it also formed other secondary structures, such as bend and turn or random coil conformation.

We also performed the ϕ and ψ angle analysis. Extended Data Table 3 shows the average ϕ and ψ values in
30 independent 1.2 μ s MD simulations for WT, pT34, and pY38. We found that there were differences in

standard deviations (SD), but the differences in the average ϕ and ψ values were smaller. Therefore, we
 consider that the secondary structure content using the DSSP algorithm² is a more comprehensive way to
 assess the disruption of the $\alpha 1$ helix, rather than the ϕ and ψ analysis. These data are shown below and are
 discussed and included in the manuscript (lines 123-125) as Extended Data Table 3.

Res.	WT				pT34				pY38			
	Ave. Phi	SD(Phi)	Ave. Psi	SD(Psi)	Ave. Phi	SD(Phi)	Ave. Psi	SD(Psi)	Ave. Phi	SD(Phi)	Ave. Psi	SD(Psi)
L15	-64.8	9.6	-34.1	10.5	-65.5	9.8	-34.4	10.6	-67.7	10.2	-33.4	11.2
K16	-68.5	10.1	-41.7	8.9	-68.1	9.9	-41.7	9.1	-67.1	10.6	-38.8	9.7
F17	-68.8	9.2	-35.4	8.1	-67.7	9.4	-36.2	8.6	-66.5	10.6	-40.5	10.5
E18	-62.7	8.2	-45.8	7.8	-62.6	8.3	-47.6	8.4	-60.5	8.9	-42.4	10.4
K19	-61.7	8.2	-39.1	8.6	-61.8	8.6	-38.2	9.3	-63.9	10.7	-36.5	11.5
K20	-63.9	8.4	-45.4	8.6	-64.3	12.2	-48.6	10.4	-67.6	15.6	-43.7	10.6
F21	-61.7	9.0	-48.6	7.8	-61.4	9.7	-45.9	9.3	-67.4	16.4	-38.0	13.8
Q22	-65.2	8.4	-35.7	9.2	-65.1	9.1	-36.3	10.4	-63.3	8.6	-36.1	10.0
S23	-68.3	9.3	-38.5	10.3	-68.8	9.7	-39.6	10.4	-69.1	10.4	-35.7	11.4
E24	-71.0	9.8	-37.3	10.2	-70.0	11.6	-36.0	14.1	-70.0	11.4	-38.0	23.6
L25	-64.7	8.9	-33.8	13.9	-66.5	12.3	-37.2	14.2	-65.9	11.2	-33.4	22.9

b. Fig1b – what structure is shown. What is starting structure for these trajectories? Were hydrogens used?

Ans: The structure in Fig1b is the WT structure we determined using X-ray crystallography in this study.
 As we mentioned in the methods section, the systems for MD simulations were constructed using the WT
 structure of Fis1 (residue 1-123) to which hydrogens were added. Protonation states were assigned using
 Adaptive Poisson-Boltzmann Solver (APBS)- PDB2PQR, which is a module solvation force library
 package with pH 7.5. We added the statement about the protonation state assignments in the Methods
 following new lines 521-525:

The systems for MD simulations were constructed using the wild-type structure of Fis1 (residue 1-123)
 revealed by X-ray crystallography published in this study. The structures of pThr34 and pTyr38 were
 modeled using molecular operating environment (MOE; Chemical Computing Group, Montreal, Canada).
 Protonation states were determined by Adaptive Poisson-Boltzmann Solver (APBS)- PDB2PQR which is
 a module solvation force library package with pH 7.5³.

 c. Figure 3 shows MD simulations and chemical modifications with CPM that show Cys41 is more solvent
 accessible upon PO4 of T34 or Y38. A minor discrepancy exists between the MD simulations with T34 v
 Y38 and the chemical modification expts that show essentially no difference. A time course might resolve
 this. What temp was the chemical modifications done at? If 25C then question is whether WT is modified
 at physiological temps.

Ans: We agree that there are some differences in the Cys41 exposure in MD simulations in the
 phosphorylated forms of Thr34 and Tyr38 but there is no major difference in the amount of chemical
 modification after 30 min incubation of the protein with the maleimide-based probe. It is possible that if
 the measurements were done at shorter time points, there could be some differences between the two
 phosphorylation mimic mutants. However, we believe that these experiments do not add much insights and
 change the conclusions of the paper. We thank the reviewer for pointing this out and we have now discussed
 this discrepancy across these two methods in the paper.

The chemical modifications were conducted at 25°C. At 25°C, the chemical modification in WT was about
 8% of the phosphorylation mimic mutant. This clearly suggests that the exposure of cysteine is different in
 the WT compared to the phosphorylation-mimic mutants. At higher temperatures, it can be presumed that
 the protein is more dynamic and there might be more chemical modification in WT, but we also expect the

chemical modification to be even higher/faster in the phosphorylation-mimic mutant. We believe that these
questions are interesting but they don't add much insights to the paper.

325 d. Line 112 “36 μ s long MD simulations each” is potentially misleading.

Ans: We agree and have modified it as follows:

“We calculated the average propensity of residues in α 1 helix (aa 15-25) to form α -helical structure using
the DSSP algorithm in 30 independent 1.2 μ s-MD simulations for each of WT, pThr34, and pTyr38.”

e. Line 149 “Fis1 is activated under oxidative stress” is missing a logical step and would benefit from a bit
more detail that included the oxidative stress conditions in the prior work (LPS) used to drive
phosphorylation.

Ans: Per this comment, we now wrote: “Additionally, Fis1 has been shown to be activated under oxidative
stress induced by cisplatin⁴, lipopolysaccharide⁴, rotenone⁴ or H₂O₂⁵ treatments in cells in culture and
oxidizable cysteine residues are well known to act as redox sensors.”

f. Consider replacing Figure 3b with a plot of the average SASA \pm sd that is reported in the text. This would
be far easier to see. For this a p-value should be reported as well. If the authors prefer the current
representation then the the figure legend should indicate what time interval is shown i.e. is the SASA data
is calculated for each ns of the trajectory

Ans: We prefer the current representation. However, as you commented, there were no descriptions of time
intervals in the figure, so we agree that it was an imprecise representation. We therefore revised the figure
to reflect your comment and added the average SASAs and standard deviations in the caption.

Original lines 903-904:

b, Solvent accessible surface area (SASA) of Cys41 (obtained from molecular dynamics simulations) for
Fis1 (green), phosphorylated Thr34 Fis1 (pThr34; blue) and phosphorylated Tyr38 Fis1 (pTyr38; red) are
shown. Each box represents independent 1.2 μ s simulation. The average SASAs of Cys41 in WT, pThr34,
and pTyr38 calculated from the 30 independent 1.2 μ s MD simulations each were 12.5 \pm 8.5 \AA^2 , 13.9 \pm 10.0
\AA^2 , and 29.2 \pm 14.7 \AA^2 , respectively.

351 g. Fig4a -consider placing residues 1-32 as dashed lines to indicate to the reader that no density was
352 observed for these residues.

Ans: Thank you for the comment. Now we have modified the figure.

355 h. Line 891 consider replacing “shown” with “modeled”

Ans: Corrected accordingly.

i. Several of the PAGE images shown are referred to as “partial native PAGE” although the methods
indicates that this was SDS-PAGE, which is non-native. This needs to be clarified. Do the authors mean
non-reducing SDS-PAGE?

Ans: Reducing agent wasn't used. Also, the samples weren't boiled prior to loading in a SDS-PAGE gel.
This method was described previously as partial native SDS-PAGE by Hwang et al. We have added this
clarification to the methods and the following reference.

Hwang, S. et al. Correcting glucose-6-phosphate dehydrogenase deficiency with a small-molecule activator.
*Nat. Commun.* **9**, 4045 (2018).

j. Line 218 "yeast Fis1 interaction with Caf4..... mediated by concave hydrophobic surfaces" is not quite
right as the crystal structure of yeast Fis1 with Caf4 (2pqr.pdb) shows both concave and convex faces of
Fis1 mediate the interaction. This paper should be cited (PubMed: 17998537).

Ans: We have rephrased the statement to say that "the interaction require concave hydrophobic surface (not
sufficient in the case of Fis1/Caf4)." Citation to the above paper is now added.

k. Figure 5 legend should state the concentration of maleimide used. Line 245 could also state this as well.
Ans: Now added.

376 l. Fig 5c – given the set up in Fig 5a and the manner in which the assay is presented in the text, Figure 5C
may be better presented as a decrease in fluorescence as opposed to % inhibition.

Ans: Our goal in Fig 5a was to show that the different constructs of Fis1 can produce fluorescence when
incubated with CPM molecule and this fluorescence can be inhibited by preincubating the protein with
unlabeled maleimide. However, the maximal fluorescence are different across different constructs of Fis1
and therefore the relative binding of small molecules to different constructs is hard to compare. Thus, we
presented the data as percent inhibition in Fig. 5c.

384 m. Line 271 The argument of protein and SP11 concentration indicating a potent inhibitor could be better
explained, especially given the covalent nature. It also contradicts what is stated in the Discussion that
needs clarification as well

Ans: We have added more explanation to better discuss the potency of SP11, as follows:

"SP11 inhibits Thr34Asp Fis1 CPM fluorescence in vitro with an IC50 of 9.4 μM. However, As SP11
binding to Thr34Asp Fis1 is 1:1 and irreversible, this IC50 is underestimated, as we used 4 μM Thr34Asp
Fis1. This half inhibitory concentration of 9.4 μM at 4 μM Thr34Asp Fis1 effectively translates to 5
molecules of SP11 inhibiting 1 molecule of Fis1, indicating excellent potency."

n. Line 348 "solution SAXS in solution " should be changed to reflect what was done.

Ans: Information is added. The revised text is:

"This observation was consistent across all modeled phosphorylated Fis1 structures and phosphorylation
mimic-mutants, using different biochemical, biophysical and structural biology experiments, including MD
simulations, solution SAXS small angle X-ray scattering in solution and X-ray crystallography of wildtype
and phosphorylation mimic mutants."

o. Fig 7e – from the figure legend and methods it is unclear at what stage the cells were treated with H2O2
and for how long.

Ans: Information is added. For mitochondrial ROS measurement and mitochondrial morphology analysis,
the cells were treated with SP11 and H2O2 simultaneously for 24h. For Drp1 translocation experiment, the
cells were pre-treated with SP11 or vehicle control for 30 min. Then the cells were treated with H2O2 in the
presence of SP11 or vehicle control and incubated for additional 24h.

p. The discussion meanders a bit and could be much tighter by focusing on the discoveries made and how
it allowed the authors to devise a unique strategy to drug the "activated" form of a protein target. The
evolutionary speculation with respect to avian Fis1 is quite interesting but seems out of place and perhaps
better suited for a review.

Ans: We respectfully disagree. We mentioned avian Fis1, as Cys41 is conserved in all vertebrates except in
avians. This observation tied to their cellular physiology provides strong indication of the potential role of
Cys41 in Fis1 in driving the effect of oxidative stress on mitochondrial structure and function. We feel that
this observation is worthwhile discussing to encourage further investigation of hypothesis implicating Fis1
in ageing/lifespan.

q. Many experimental details are missing with respect to concentrations, incubation times, amounts of cells
used, etc. Figure legends need updating with number of experiments and whether uncertainties are std dev
or not. Gels need to be quantified and number of replicates indicated.

Ans: All these are now added.

r. What cloning artifact does the protease cleavage leave. This should be stated and the length of the
construct in the methods.

Ans: We were mindful that most of the changes are happening/ anticipated to happen in the N-terminus of
Fis1 and therefore we opted to engineer the hexa-histidine tag and the protease cleavage site on the C-
terminus of the recombinant Fis1 construct. As protease cleavage site is located on the C-terminus, we don't
anticipate this to add significant artifact to our observations which are mostly focused on the N-terminal
region.

The Fis1 sequences used in the experiments are now added as Supplementary Fig. 7.

432 s. Fis1 has been reported to act via inhibition of the fusion machinery (PMID: 30842096), and since no data
in the submitted manuscript eliminates this possibility, it should be incorporated into their model and
discussion.

Ans: This discussion is now added. Specifically:

“Our study focusses on the role of activated Fis1 in causing excessive mitochondrial fission and dysfunction
through increased Drp1 recruitment to mitochondria. In addition to promoting fission through Drp1, it is
possible that the inhibition of fusion machinery by activated Fis1 also contributes to the fragmented
mitochondrial phenotype under cellular stress. Previously, it was shown that Fis1 inhibits proteins involved
in mitochondrial fusion such as Mfn1/2 and Opa1⁶ and this inhibitory effect of activated Fis1 needs to be
investigated.”

Blake Hill

Dr. Hill, Thank you for the very constructive and detailed criticism that helped improve our manuscript.

**Reviewer #2:**

Review of NCOMMS-23-64230-T: A hidden cysteine in Fis1 targeted to prevent pathological
mitochondrial fission and dysfunction by Pokhrel et al.

This is an interesting manuscript that characterizes novel conformational changes that permit redox
sensitive crosslinking of Fis1, a receptor of Drp1 that promotes mitochondrial fission. This dimerization is
observed in new crystal structures and the authors used MD to show that phosphorylation at specific sites
can destabilize an N-terminal helix to allow for this crosslinking to occur. The studies are rigorous and the
data novel. Overall, the most compelling part of the story is the identification of a novel Fis1 dimerization
that could promote organelle fission in a redox-sensitive manner. The major limitation of the paper is the
lack of direct evidence that this crosslink exists in cells. The in vitro work is solid and shows that the reactive
Cys is engaged in dimerization. The use of a small molecule, SP11, inhibited this linkage through direct
interactions with the Cys. Appropriate controls were used to show the selectivity of the molecule for this
site, but these same controls were not pursued in cellular experiments that would definitively highlight the
role of this linkage in promoting mitochondrial fission.

Otherwise, the flow of the paper is a little choppy and some suggestions are provided to help with any
revisions.

1. The introduction is very terse. While I appreciate being direct and to the point, additional information
could and should be provided to highlight the roles of Drp1 and Fis1 and expand on what is known about
the Drp1-Fis1 interaction. This can highlight the gaps in knowledge since this paper seeks to modulate this
interaction. I especially would recommend incorporating some discussion of previous structural studies
with the human Fis1 to frame the impact of this study.

*Ans: We thank the reviewer for the suggestion and have modified the introduction accordingly. We added
the following:*

*“The cytosolic domain of Fis1 is composed of 6 tight α -helices with two anti-parallel tetratricopeptide*
*repeat (TPR) motifs forming a concave hydrophobic surface. Such large hydrophobic patches are critical*
*for protein-protein interactions. One way to regulate the interactions of such proteins is by autoinhibition,*
*in which part of the protein folds in and occludes the hydrophobic surface, which then folds away in*
*response to certain signals rendering the protein competent for interactions. In yeast, the N-terminal region*
*of Fis1 was shown to be autoinhibitory as it folds inward to occupy the concave hydrophobic surface critical*
*for interaction with Drp1; removal of the first 16 residues in N-terminus increases Drp1 binding. Similarly,*
*a peptide, identified by phage display, binds human Fis1 only after the N terminal α 1-helix was removed.*
*However, the N-terminal α 1-helix was previously shown to be indispensable for Fis1-mediated*
*mitochondrial fission in mammalian cells. This divergence in the function of the N-terminal region of yeast*
*and human Fis1 despite their structural similarity indicates potential additional regulatory role of the N-*
*terminal region in human Fis1. The structural basis of human Fis1 activation, subsequent Drp1 recruitment*
*to mitochondria and regulation of mitochondrial fission under cellular stress is unknown.”*

2. A representative movie comparing the fluctuations observed by MD in Figure 1 would be helpful. This
would highlight the extent to which these changes are affecting the helical structure and movement.

*Ans: Per the reviewer’s suggestion, we added Extended data Movie 1 to visually highlight the dynamic
nature of phosphorylated Fis1. We also modified the text in lines around 119-125 as follows:*

*“We calculated the average propensity of residues in α 1 helix (aa 15-25) to form α -helical structure using*
*the DSSP algorithm in 30 independent 1.2 μ s-MD simulations for each of WT, pThr34, and pTyr38.*
*Compared to unphosphorylated Fis1 (WT), the propensity of amino acid residues 15-25 to form α -helical*
*structure in pThr34 and pTyr38 were lower by 0.19-2.22% and 1.91-6.44%, respectively (Fig. 1c, Extended*
*Data Fig. 1, Extended Data Movie 1). We also performed the ϕ and ψ angle analysis in these simulations.*
*We found that there were differences in standard deviations, but the differences in the average ϕ and ψ*
*values were smaller (Extended Data Table 3).”*

3. The crystallography data are nice, but the absence of something (i.e. α 1) is not proof of disorder. The

SAXS analysis is nice, but NMR or some spectroscopy method would seem more appropriate. Others have
had success with NMR of Fis1, and they did not see the movements in alpha1, but they could discern the
flexibility in the N-terminal IDR.

Ans: Indeed, every technique comes with limitations, and it is important to verify the result with
complementary techniques. We have shown evidence of α -1 helix unfolding using different complementary
techniques, including MD simulations, small angle X-ray scattering experiment, X-ray crystallography and
CPM fluorescence assay. The results are consistent across all the techniques and across all the
phosphorylation mimic mutants, strongly supporting our conclusions about the flexibility of α 1 helix.

Our crystal structure does not only show the absence of α -1 helix but also shows that there is no space for
structured α -1 helix in the covalent dimer, as this space is occupied by α -2 helix of another monomer (see
Extended Data Fig. 3). We further clarified this in our revised discussion:

“Our crystal structure not only shows the absence of α -1 helix but also shows that there is no space for
structured α -1 helix in the covalent dimer as this space is now occupied by α -2 helix of another monomer
(Extended Data Fig. 3). It can be argued that absence of structure may not always be proof of disorder, but
in our context that seems to be the only explanation. This observation was consistent across all modeled
phosphorylated Fis1 structures and phosphorylation mimic-mutants, using different biochemical,
biophysical and structural biology experiments, including MD simulations, solution SAXS small angle X-
ray scattering in solution and X-ray crystallography of wildtype and phosphorylation mimic mutants.”

4. Related to this previous point... the early mention highlighting that alpha1 is missing due to disorder
seems a misleading when it is revealed that the construct formed a disulfide linked dimer in the lattice. This
dimerization would seem incompatible with the formation of alpha1 (see Extended Fig 2). But the
underlying cause for the loss of alpha1 could be stabilization of the crosslinked dimer. Do the densities
presented in Fig 2 have the dimer pair density subtracted? I would highlight the dimerization earlier in the
presentation of the structure and highlight the incompatibility with formation of the alpha1 helix when the
protein forms this dimer. Currently, this gets lost and buried for me.

Ans: We disagree with the reviewer that the underlying cause of the loss of α -1 helix is due to the
stabilization of the crosslinked dimer because we see the increased fluctuation and unfolding propensity
even in monomers in the MD simulations, small angle X-ray scattering experiment and CPM fluorescence.
We first show that the N-terminus is dynamic, and it gets disordered upon perturbation. As a consequence,
the critical Cys41 gets exposed and is capable of forming a disulfide bond with another monomer to form
a covalent dimer; that is the reason for the order of data presentation that we used.

5. While the phosphomimetics offer compelling evidence for exposing Cys41, in vitro phosphorylation
would provide validation that activation of the protein enhances Cys41 accessibility. Previous groups used
this with Met to phosphorylate the protein in vitro at Y38, so this may be feasible.

Ans: Producing enough homogeneously phosphorylated Fis1 protein for biophysical measurement is not
feasible and phosphorylation mimic mutants recapitulating the effect of phosphorylation of different
proteins have been extensively described in the literature. Specific to Fis1, three studies^{4,5,7} independently
demonstrated that phosphorylation mimic mutants at Thr34 and Tyr38 position can faithfully recapitulate
the effect of phosphorylation in cells in culture. We now have referenced these studies to support our use
of the phosphomimetic mutants in our biophysical study; Lines 77-81.

6. The protein purification was done in the absence of any reducing agent, which may explain how the
phosphomimetic proteins were able to form disulfide linked dimers in the crystal. But previous structural
work with Fis1 were not oxidized, and this may explain why the destabilization of alpha 1 was observed.
Can these dimers be induced after purifying reduced protein? How interchangeable are these interactions?

Ans: Indeed, dimers can be induced after purifying reduced protein. As per the reviewer's suggestion, we
immobilized the same amounts of WT and phosphorylation mimic mutant proteins and incubated in a buffer
with 10 mM 2-ME for an hour (to break all pre-existing covalent dimers), washed away the 2-ME incubated
at room temperature and observed dimerization over time. We observed that WT Fis1 has very low

propensity to form covalent dimers compared to the phosphorylation mimic mutants (see new data provided
below). These data are included in the manuscript as Extended Data Fig. 5.

7. Related to the above point... Can these dimers be captured from endogenous Fis1 in the cell? Can they
be captured to show that oxidation can lead to similar dimers without forcing them in vitro? The lack of
cellular studies to identify the disulfide crosslink is a major limitation. The authors used magnetic beads to
force His-tagged construct into close proximity. But could this cross-link be seen in cells? Organlle
association could be driven by a number of independent factors. In addition to H₂O₂, did the authors used
the phosphomimetic proteins to enrich for crosslinks that could be captured? Without this evidence, these
data could represent an in vitro phenomenon.

Ans: We haven't been able to capture covalent Fis1 dimers from the cells, likely because the cytosol is
reducing, and these disulfide bonds are reversible⁸.

Using an alternative approach to establish the biological significance of Cys41-mediated dimerization cells
under oxidative stress, we generated a Cys41Ser knock in M17 cells. Relative to WT M17, the Cys41Ser
cells have reduced sensitivity to oxidative stress as seen in the MitoSOX assay. These data, combined with
*in vitro* data on Cys41-mediated covalent dimerization, reasonably establish that Cys41-mediated Fis1
dimerization is critical in Fis1-mediated mitochondrial fragmentation and dysfunction under stress (now
provided as Fig. 7f).

8. It is unclear to me where the SP22 analogue came from or how it serves as a control for the SP11 experiments. It appears to behave similarly, though maybe with less potency.

Ans: We used SP22, a molecule with the same phenothiazine core scaffold as SP11, but a different covalent warhead to rule out that the binding of SP11 to Thr34Asp Fis1 is not an artifact of the warhead. This control establishes that the binding of SP11 to Thr34Asp Fis1 is driven by the binding of phenothiazine core to the binding pocket of Thr34Asp Fis1 and not just due to the intrinsic reactivity of the chloroacetamide warhead. This clarification is now added to the manuscript (lines 314-318).

9. Do the SAXS experiments have the resolution to reliably identify a conformational change in a single alpha helix in this small protein? The envelope seems similar, so I am not sure where the difference is predicted. Also, P2 symmetry was implemented. Were densities examined without symmetry applied to see what the resulting maps looked like? It may be beneficial to test this.

Ans: We do not think current state-of-the-art SAXS technology has the resolution to reliably identify a conformational change such as a small kink in a single alpha helix in a small protein. However, if the conformational change is such that an alpha helix is completely disordered causing the shape and apparent volume of the molecule to change, it can be detected using SAXS as we have shown with the WT and phosphorylation-mimic mutants of Fis1 (Fig. 2b). We have used SAXS in our study to answer two questions: (1) Difference in apparent shape and volume of Fis1 in the WT and phosphorylation forms due to alpha-1 helix unfolding. (2) Distinguish the shape of WT dimer in solution to determine whether it is similar to the crystallographic dimer or the phosphorylation mimic mutant covalent dimer. We conclude that the data we have obtained from SAXS addressed both questions reasonably well. Furthermore, the results from SAXS were substantiated using orthogonal experiments, such as CPM fluorescence, reducing/non-reducing SDS-PAGE, X-ray crystallography, etc.

To answer your second question whether the shape of the WT dimer in solution is similar to the crystallographic dimer or the phosphorylation-mimic mutant covalent dimer, we collected experimental SAXS data for WT dimer (Kratky plot – blue circles in Fig 4e) and generated a theoretical Kratky plot blue (solid line Fig. 4e) using the crystallographic non-covalent dimer. When these two plots were compared, we clearly see that the theoretical plot does not fit the experimental plot, suggesting that the shape of Fis1 dimer in solution is different than from the crystal structure. Then, we modeled a covalent WT Fis1 dimer using Thr34Asp Fis1 covalent dimer as a template and modeled in the missing N-terminus. We then used this structure to generate a theoretical Kratky plot (solid line Fig. 4f). This theoretical Kratky plot perfectly fitted the experimental Kratky plot of WT Fis1 dimer (blue circles in Fig 4e and f; both are same), suggesting that the structure of Fis1 dimer in solution is similar to modeled Fis1 covalent dimer. We generated an *ab initio* envelope, using experimental SAXS data (shown in Fig. 4f) and found that one of the models of Fis1 covalent dimer fits that of the *ab initio* envelope developed using experimental SAXS data (Fig. 4f). Here, the SAXS experiment clearly distinguished the shape of WT in solution. The SAXS analysis and modeling were used to confirm the overall shape of the dimer in solution and not to

609 distinguish between different conformations of the α helix. The covalent nature of the WT Fis1 dimer in
solution is also confirmed by reducing vs. non-reducing SDS-PAGE.

The reviewer is correct in pointing out the P2 symmetry was implemented while generating the *ab initio*
density and envelopes. We implemented the P2 symmetry because all the structures of Fis1 dimer were in
P2 symmetry and the goal of this experiment was to fit the dimer structure with P2 symmetry into the
density.

Generally, it is advisable to incorporate available constraints, such as symmetry operation and anisometry,
during low-resolution SAXS modeling. This approach can mitigate or reduce issues arising from the
averaging of the resulting models (e.g. envelopes) that most SAXS modeling programs are employed. At
the superimposition process of the models for the averaging, the structural details far from the weighted
center or near the outer edge can be easily averaged out (although this depends on how the superimpositions
are performed). Due to the low information content of SAXS data, the enantiomorphs (chirality) of the
resulting modes are another issue in the averaging process. Since an enantiomorphous structure gives the
same scattering, the resulting individual models could contain enantiomorphous structures and additional
error may be observed in the search for enantiomorphs during the averaging. In many cases, discrepancies
observed between constrained and unconstrained models (e.g. without vs. with symmetry) may not be
indicative of the quality or reliability of the SAXS data and results, but rather may be due to unfavorable
shape in such averaging procedures. Therefore, we are hesitant to include the envelope in unconstrained
calculations (without symmetry) to avoid misinterpretation of the data quality by the reader.

As suggested by the reviewer, we examined the density without implementing the symmetry i.e. we used
P1 symmetry and did not observe significant differences in the shape of the envelope when P1 or P2
symmetry was enforced (data shown below). This point is now clarified in the manuscript; Lines 253-257.

These data are now added as Fig. 4g.

10. A major limitation in the field is the inability to reproducibly observe Drp1-Fis1 interactions with
isolated proteins. Did the authors attempt to examine the impact of Fis1 dimerization on Drp1 activity
and/or structure? This would help define the impact of this change on mitochondrial fission.

Ans: We have used several techniques such as pulldown, fluorescence polarization and X-ray
crystallography to observe Drp1-Fis1 interaction *in vitro* with purified proteins, and like other researchers
in the field, have never been able to reliably show Drp1-Fis1 interaction with the purified proteins. We
suspect that mitochondrial membrane and post translational modifications of Drp1 and/or Fis1 play a
critical role in their interaction and these factors are missing in the *in vitro* context.

11. Do we know whether the SP11 interacts with H₂O₂ directly? Addition of both in the same media may
directly lead to less oxidation if there is an interaction with H₂O₂. It would be nice to identify a control that
would be insensitive to SP11 and show that it is still leading to ROS stress when treated with both H₂O₂
and SP11. Any evidence to show that there is no interaction would suffice tbh.

Ans: We don't have a reason to believe that SP11 directly interacts with H₂O₂. In our cells in culture
experiments, SP11 is used at 250 nM and H₂O₂ is used at 50 or 100 μM and therefore the reduction in
effective H₂O₂ due to interaction/scavenging by SP11 would be negligible (even if that happens).
In response to the reviewer's comment, we made Cys41Ser knock in homozygous M17 cells that are
insensitive to SP11. But these cells already have reduced sensitivity to H₂O₂ as seen in the MitoSOX assay
and therefore the interaction of SP11 and H₂O₂ couldn't be assessed in this experiment as suggested by the
reviewer. This data are provided in Fig 7f.

Minor comments:

12. The mention of SP11 in the Abstract confusing, since this has not been defined. I would note that this
is a small molecule designed as an inhibitor for Drp1 in this section.

Ans: now corrected.

“Our discovery of a small molecule, **SP11**, that binds only to activated Fis1 by engaging Cys41...”

13. The dimerization that was compared in the paper (ref 17, line 182) was looking for a different, inhibitory
dimerization. This was identified in the Yeast Fis1 protein several years ago (Lees et al, JMB, 2012). Some
comparison with this model would help define the unique attributes of this dimerization interface through
Cys41 and highlight why this could be productive, rather than inhibitory. In fact, this comparison would
highlight regions that would be available for interactions with Drp1 directly.

Ans: The yeast Fis1 dimer is non-covalent as reported by Lees et al.¹¹, where they show that Fis1 ΔTM can
form dimers even when both cysteine residues, Cys79 and Cys87, are mutated to serine whereas human
Fis1 dimer is covalent (this study). Additionally, Fis1 in yeast is the only mitochondrial adapter that recruits
Drp1 but in humans and other mammals, Fis1 recruits Drp1 only under pathological conditions. As human
and yeast Fis1 have different basis of dimerization and physiological context, we were unable to draw
hypothesis regarding the interaction interface between Fis1 and Drp1 by comparing the human covalent
dimer structure with yeast Fis1 dimer model. However, this divergence could potentially be a basis of
evolution of new functions of Fis1 in higher organisms. A discussion of this point is now added:

“Previously, non-covalent nature of Fis1 dimer was reported in yeast where it is the only mitochondrial
adapter that recruits Drp1. However, in mammals, Fis1 recruits Drp1 only under pathological conditions.
Therefore, it is not surprising that the effect of dimerization in mammals (exclusively under pathological
conditions) and in yeast (under physiological condition) leads to different functional effects on the ability
of the protein to recruit Fis1.”

14. The legend for Extended Figure 6 explains things for SP11 instead of SP22, which the compound used
in these experiments I think.

Ans: Thank you for pointing this out. Extended Data Fig. 7 (previously Extended Data Fig. 6) characterizes
SP22. The caption is now corrected.

**Reviewer #3:**

In this study, the authors investigate Fis1 activation in response to stress through structural changes
alterations that mediate Drp1 binding on the mitochondria. This regulatory mechanism while poorly defined
is important because Fis1-Drp1 interaction mediates pathological mitochondrial fission. The N-terminal
region of Fis1 is dynamic and the author speculate that perturbations such as phosphorylation at key tyrosine
on threonine residues on the α 2-helix cause unfolding to expose the single cysteine residue C41. Oxidized
C41, presumably through sensing of oxidative stress, mediates Fis1 covalent dimerization and activation.
The authors use a combination of MD simulations, site-directed mutagenesis and structural biology to
support this intriguing regulation.

Next, the authors screen for cysteine-reactive fragments to bind and block Fis1 covalent dimerization
activation and identify a lead compound SP11, a phenothiazine-chloroacetamide, that bound C41 on
activated Fis1. The authors also test an acrylamide counterpart (SP22) that show comparable effects. Site
of binding for SP11 was confirmed by co-crystal structures and LC-MS/MS. This manuscript displays an
impressive array of diverse methods to tackle this challenging problem. The identification of covalent
inhibitors of Fis1-Drp1 protein-protein interaction (PPI) via disruption of a disulfide is important for
guiding PPI inhibitor development. Fis1 inhibitors could also have therapeutic applications.

While the studies are promising, I have a major issue with the lack of selectivity studies in cellular studies
using SP11 and the claims made with regards to Fis1 function using this compound. While I appreciate the
authors introducing a second cysteine site to counter screen for non-specific binders, these studies are still
only testing covalent binding of compounds against a single protein. Chemical proteomic methods are
available to evaluate selectivity of cysteine-binding ligands such as SP11 and authors should use these
methods to provide evidence of reasonable selectivity in SP11-treated cells. For example, alkyne-tagged
iodoacetamide is an effective probe for evaluating ligand binding to cysteines proteome-wide. Additionally,
I see no evidence for target engagement of Fis1 by SP11 in cellular studies. These studies are important for
supporting the claims that the mitochondrial phenotypes observed in cellular studies are due to disruption
of Fis1-Drp1. I suggest including a cysteine-reactive but Fis1-inactive compound as a negative control to
further support Fis1 on-target effects.

**Ans:** We agree with the concerns raised by the reviewer. To address the concern regarding target
engagement, we have added new evidence using Cys41Ser knock in homozygous M17 cells. When these
cells are treated with H₂O₂ they have reduced sensitivity to oxidative stress and treatment with SP11 has no
effect on MitoSOX signal unlike cells expressing WT Fis1. Similar trend was observed in H₂O₂ induced
mitochondrial localization of Drp1 in cells expressing WT and Cys41Ser Fis1 when treated with SP11.
These experiments establish that the effect of SP11 treatment is through engagement with Fis1 at Cys41.
These data are provided below and included in the manuscript as Fig. 7f.

Furthermore, we find that SP11 binding to the same reactive Cys41 in Fis1 is abolished when Tyr38 was
mutated to glutamic acid (Fig. 6e). This is a good indication that the SP11 is not a promiscuous binder.
However, as the reviewer mentioned, the selectivity of SP11 against all other cysteines in the proteome
needs to be established; this limitation is now discussed in the discussion section:

“While we have used counter screen construct of Fis1, and Tyr38Glu Fis1 mutant to show SP11 binding
isn’t promiscuous and Cys41Ser Fis1 homozygous cells to demonstrate that the biological activity of SP11
is mediated by binding to Cys41 on Fis1, the selectivity of SP11 against any other cysteine in the proteome
needs to be established. Further chemi-proteomics analysis and medicinal chemistry efforts to optimize
SP11 for target selectivity will identify lead compounds to test in preclinical animal model of diseases.”

Additional considerations:

- Figure 4b – to further support covalent dimers in solution, can authors add BME to disrupt and evaluate
by native gel?

Ans: These data are now provided. The last 3 lanes in new Fig. 4b show the effect of BME on the dimers.
The dimers are completely lost in the presence of BME in all three constructs, suggesting that the Fis1
dimers are covalent dimers mediated by Cys41, the single cysteine in Fis1. This is now clarified; lines 220-
221.

- Claim of SP11 having drug-like characteristics in the discussion section is a bit overstated given there are
no data supporting this statement.

Ans: There are over a dozen of FDA approved drugs based on phenothiazine moiety. These drugs are known
to have great pharmacological properties, such as being orally bioavailable with longer half-lives and blood
brain barrier permeability. As SP11 contains core phenothiazine moiety, we edited the speculation of good
drug-like characteristics of SP11, per the reviewer’s comment.

Specifically:

“Many phenothiazine-containing drugs are orally bioavailable, have longer half-lives and are blood brain
barrier permeable and thus, it is likely that SP11 also has good drug-like properties.”

- Authors claim SP11 has excellent potency in the discussion. This claim should be supported by a
measurement of potency for covalent inhibitors. For example, a measure of K_{inact}/K_I would be appropriate.

Ans: We appreciate reviewer’s suggestion regarding the measurement of K_{inact}/K_I to determine the
potency of SP11. However, Fis1 is a structural protein without any known enzymatic activity and there are
no direct in vitro functional assays to measure inhibition. Therefore, we are unable to measure the K_I . Our
assay can only measure the covalent bond formation, K_{inact} component, but not K_{inact}/K_I . This discussion
is now added. Specifically:

“However, we were unable to measure K_i and determine K_{inact}/K_i for SP11 as Fis1 is a structural protein
without any known enzymatic activity; there are no direct in vitro functional assays to measure Fis1
inhibition.”

- Figure 6A: MS/MS spectra shows many peaks that are not annotated. Can the authors explain? How
confident are authors in the identification? Other metrics used to further support quality of the modified
peptide?

Ans: The goal of MS/MS spectra was to unequivocally establish that SP11 binds to Thr34Asp Fis1 at Cys41
position. We have clear, sufficient, and high-intensity peaks for peptides in the provided spectrum and we
could identify peak corresponding to peptide containing SP11-modified Cys residue. This experiment was
done at a commercial research organization (BGI Global Genomic Services) that was blinded to the
experimental conditions and did not have any interest in the results. Based on their data, they independently
came to this conclusion that was in accordance with our hypothesis. As Orbitrap is a high-resolution

instrument, it is not uncommon to find background noise with high intensity and we speculate this to be the
reason behind the unassigned peaks.

We have now added new intact mass spec data from the same sample used in peptide mapping. This intact
mass spec showed that Thr34Asp Fis1 + SP11 sample has a new peak shifted by 310 Da (weight of SP11
adduct) compared to untreated Thr34Asp Fis1, suggesting that there is only one SP11 bound to Thr34Asp
Fis1. These data are provided below and included in the manuscript as Supplementary Fig. 3.

These two pieces of data taken together provides unequivocal evidence of SP11 binding to Cys41 in Fis1
(which is the only Cys residue in Fis1). Note also that we have several lines of evidence including X-ray
crystallography of the Thr34Asp Fis1-SP11 complex, CPM fluorescence assay, inhibition of dimerization
assay that support this result from MS/MS spectra.

786 References

- 1. Nolden, K.A., Harwig, M.C. & Hill, R.B. Human Fis1 directly interacts with Drp1 in an
evolutionarily conserved manner to promote mitochondrial fission. *J Biol Chem* **299**, 105380
(2023).
- 2. Kabsch, W. & Sander, C. Dictionary of protein secondary structure: pattern recognition of
hydrogen-bonded and geometrical features. *Biopolymers* **22**, 2577-637 (1983).
- 3. Jurrus, E. et al. Improvements to the APBS biomolecular solvation software suite. *Protein Sci* **27**,
112-128 (2018).

- 4. Wang, S. et al. DNA-PKcs interacts with and phosphorylates Fis1 to induce mitochondrial
fragmentation in tubular cells during acute kidney injury. *Sci Signal* **15**, eabh1121 (2022).
- 5. Zou, R. et al. Empagliflozin attenuates cardiac microvascular ischemia/reperfusion injury through
improving mitochondrial homeostasis. *Cardiovasc Diabetol* **21**, 106 (2022).
- 6. Yu, R., Jin, S.B., Lendahl, U., Nister, M. & Zhao, J. Human Fis1 regulates mitochondrial dynamics
through inhibition of the fusion machinery. *EMBO J* **38**(2019).
- 7. Yu, Y. et al. Fis1 phosphorylation by Met promotes mitochondrial fission and hepatocellular
carcinoma metastasis. *Signal Transduct Target Ther* **6**, 401 (2021).
- 8. Cumming, R.C. et al. Protein disulfide bond formation in the cytoplasm during oxidative stress. *J*
*Biol Chem* **279**, 21749-58 (2004).
- 9. Qi, X., Qvit, N., Su, Y.C. & Mochly-Rosen, D. A novel Drp1 inhibitor diminishes aberrant
mitochondrial fission and neurotoxicity. *J Cell Sci* **126**, 789-802 (2013).
- 10. Disatnik, M.H. et al. Acute inhibition of excessive mitochondrial fission after myocardial infarction
prevents long-term cardiac dysfunction. *J Am Heart Assoc* **2**, e000461 (2013).
- 11. Lees, J.P. et al. A designed point mutant in Fis1 disrupts dimerization and mitochondrial fission. *J*
*Mol Biol* **423**, 143-58 (2012).

REVIEWER COMMENTS

Reviewer #1:

The revised manuscript entitled “A hidden cysteine in Fis1 targeted to prevent excessive mitochondrial fission and dysfunction under oxidative stress” is significantly improved and many of the results interpreted appropriately and new data has been added to address the majority of referees’ concerns, although a few lingering issues exist:

1. Specificity of SP11.

Previously, the authors had reported that treatment of HK-2 cells with H₂O₂ increased mitochondrial fragmentation, mitochondrial reactive oxygen species (mtROS, as measured using the MitoSOX probe), and Drp1 recruitment to mitochondria. The authors went on to show that pre-treatment of the SP11 compound prevented the H₂O₂-induced effects. These data were presented in the originally submitted manuscript, but did not address whether SP11 was acting by targeting Fis1. Given the reactive nature of SP11, this is an important issue raised by referees. To address this, the authors used a Cys41Ser knock-in cell line and new data (Fig 7f) show no increase in mtROS upon treatment with H₂O₂, which supports the claim that mtROS increase in WT is generated by Fis1 oxidation at Cys41. Treatment with the SP11 compound under these conditions has no effect suggesting that off-target effects of SP11 are not responsible for the decrease in mtROS generation. However, mitochondrial morphology and Drp1 recruitment were not examined in the new cell line, and would strongly support their claims. The rebuttal mentions that Drp1 recruitment was examined in this new cell line...perhaps the data were mistakenly not included?

Ans: Unfortunately, Be(2)-M17 cells are not suitable for mitochondrial imaging since the cells are very small, with a very small cytosolic (non-nuclear) area. We added a new experiment in which WT and Cys41Ser Fis1 expressing plasmids were transfected into Fis1 KO MEFs, and mitochondrial morphology was studied in different treatment conditions. (See below and new Fig. 7g). In Fis1 KO MEFs transfected with plasmid expressing WT Fis1, 50 μ M H₂O₂ treatment caused mitochondrial fragmentation; almost 50% of the cells showed fragmented mitochondrial phenotype and pretreatment with 250 nM SP11 brought mitochondrial fragmentation to 15% (the same as basal fragmentation in Fis1KO cells or cells expressing Cys41Ser Fis1). Importantly, H₂O₂ treatment of Fis1 KO MEFs or Fis1 KO MEFs transfected with plasmid expressing Cys41Ser Fis1 did not show fragmented mitochondrial phenotype and pretreatment with SP11 had no effect on H₂O₂ treatment. These results are consistent with mitoSOX data in M17 cells (Fig. 7f).

2. Helix 1 unfolding

The model the authors propose is that phosphorylation of either Thr34 or Tyr38 unfolds helix 1, exposing Cys41, causing disulfide-mediated dimerization of Fis1 (Cys41 is the sole Cys in Fis1), which allows for enhanced Drp1 recruitment. The data largely support the model except for helix 1 unfolding and Fis1 dimer, not monomer, binding Drp1 (point #3 below). The evidence that helix 1 is unfolded is not compelling, and it is not necessary. The unfolding interpretation is based on 3 observations: (1) MD simulations (2) lack of density in x-ray crystallography, and (3) SAXS data. The MD does show local unfolding is more possible in the pT34 and pY38 constructs compared to wild type, however, the phi/psi analyses and the DSSP analyses suggest a modest unfolding is possible, and don't support the claims that helix 1 unfolds. As Referee 2 pointed out, absence of density in x-ray crystallography is consistent with unfolding but also consistent with other types of disorder less severe. SAXS data has the possibility to resolve these issues and the $P(r)$ plots are nearly superimposable with a longer tail for WT and T34D than Y38E (reaches zero at $\sim 60\text{\AA}$ for WT and $\sim 56\text{\AA}$ for Y38E). This is opposite one would expect for helix 1 unfolding in the phosphomimetics compared to wild type. Additionally, the AUC under the curve from the Kratky plot is consistent with Y38E being more compact as it is the most symmetric curve with the lowest AUC value under 4. The Porod volume is larger for Y38E and this likely is the basis for their interpretation. Perhaps plotting their existing data as $q^3 I(q)$ vs q^3 or $q^4 I(q)$ vs q^4 could address this, but to me these data are indistinguishable. My suggestion is that the authors consider replacing the occurrences of helix 1 unfolding with a conformational change involving helix 1 (that exposes Cys41). This is exactly what Extended Data Figure 2 shows, and is the most consistent interpretation of their data. If the authors insist on interpreting their data as unfolded then CD, or better, NMR, should be provided to support the claims. Alternatively, limited proteolysis could be used to probe the lability of the N-terminal region of WT vs phosphomimetics.

Ans: We thank for the reviewer's insightful suggestion and now corrected the text throughout to say a conformational change involving helix 1 that exposes Cys41, instead of stating that helix 1 unfolds.

3. Is Fis1 dimerization necessary for Drp1 interaction

The manuscript would be improved mechanistically if the authors could determine whether Fis1 dimer directly interacts with Drp1. This group previously published in Nature Communications using recombinant Drp1 and so they can produce both proteins and test for binding. I respect that they want to reserve this for future work, but it would improve their model. I note that their statement to Referee #3 that prior work has not established a Fis1-Drp1 direct interaction neglects our work (see PMID:37866629), which I pointed out in my earlier review. The authors are correct in that other groups (Jason Mears and Jodi Nunnari, personal communication) have not observed a direct interaction, but I think this is because it is autoinhibited by the N-terminal region as we stated in our JBC papers on this subject (only one of which was referenced, PMID: 37866629 and 37777154 are not). Given the authors model in which the N-terminal region prevents exposure of Cys41, it seems appropriate that this prior work incorporated in their discussion.

Ans: We have added new data to show that Cys41, which we found in vitro to mediate Fis1 dimerization, is required for Drp1 recruitment to mitochondria. We showed that H_2O_2 induced Drp1 mitochondrial translocation only occurs in Fis1 KO MEFs transfected with plasmid expressing WT Fis1, but not Cys41Ser Fis1 (See below and new Extended Data Fig. 9).

This result establishes that Cys41, which *in vitro* we showed to mediate Fis1 dimerization, is required for Drp1 translocation to mitochondria under oxidative stress. However, as correctly pointed out by the reviewer, we did not demonstrate direct physical interaction of Drp1 with Fis1 dimers. Prior work by Dr. Hill's lab supports our interpretation that N-terminal helix of Fis1 is autoinhibitory and conformational change of this helix causes exposure of Cys41 leading to Fis1 dimerization and Drp1 recruitment. These new data have been added, and all the prior relevant work has been referenced and discussed as suggested. We apologize for unintentional failure in referencing these studies in our previous draft.

The added section is as follows (also see response to comment 4 below):

“In cells expressing Fis1 in which Cys41 was mutated to Serine, H₂O₂-induced Drp1 recruitment to mitochondria, mitochondrial fragmentation and elevation of mitoROS did not occur and treatment with SP11, which binds to Cys41 in WT Fis1, consistently phenocopied the cell expressing Fis1 that lacks Cys41 and Fis1 knock out cells. Although a direct physical interaction between Drp1 and Fis1 dimer was not demonstrated in this *in culture* study, these results establish that Cys41, which *in vitro* we showed to mediate Fis1 dimerization, is required for Drp1 translocation to mitochondria under oxidative stress.”

4. Discussion

The revised discussion deleted the first paragraph that placed their work in context of earlier work (including ours. Ref 18) showing an important role for the Fis1 N-terminal region. It is unclear why this paragraph was deleted. As noted above (#3), the authors should also incorporate PMID: 37866629 and 37777154 in their discussion given the important role of the N-terminus to their model.

Ans: This section of the manuscript was moved from discussion section to introduction section as per Reviewer 2's suggestion. Now we have added a new section to discuss the important role of the N-terminus. The added section is as follows:

“How does Fis1 recruit Drp1 to mitochondria only during cellular stress and not under basal conditions? Previous data shows that the N terminus of Fis1 inhibits it: autoinhibition of Fis1 by the inwardly folded N-terminal region was shown in yeast³ [PMID: 17884824]. More recently, similar regulatory nature of N-terminal region of human Fis1 was demonstrated, using computational, biophysical and functional methods^{1,2,4} [PMID: 36272645; 37777154; 37866629]. These studies showed that the N-terminal region of Fis1 is flexible and can adopt different conformations and hence can regulate the activity of Fis1. In this study, we identified how a conformational change in the N terminus activates Fis1 to bind Drp1.”

Minor points-

5. Using the terminology partial native SDS-PAGE is confusing and inaccurate. SDS is a denaturant and it implies something different than what was done, despite Hwang et al usage. As described this is non-reducing SDS-PAGE in which the samples are not boiled, a common practice in reducing and non-reducing SDS-PAGE. Hwang et al is the only reference I could find that has this terminology and this reference is from the same lab, so I do not see this as a generally accepted terminology.

Ans: Now corrected to say “non-reducing non-heat denatured SDS-PAGE” instead of “partial native SDS-PAGE” terminology.

6. Line 1010 – porod should be capitalized

Ans: Now corrected.

7. Line 572 – percent is misspelled. The document should be checked for other misspellings.

Ans: Now corrected.

8. CPM Assay – the descriptions of concentrations in the CPM assay are confusing. What is meant by test molecule? Stating whether this is CPM itself or the protein being tested would clarify.

Ans: Now corrected to say test small molecule with cysteine reactive covalent warhead.

9. Reporting an IC50 is unclear. Is this from equilibrium measurements? i.e. waiting ~62 hours? If not, then it should be reported as an “apparent IC50”.

Ans: Now corrected.

Reviewer #2

The authors did a thorough job in addressing the reviewers comments throughout, and I appreciate the effort that went into the additional data that is now presented.

The changes in the text are thoughtful and largely address the reviewers' concerns.

There are inherent limitations with this system, but this work represents an important advance for the field. I am in favor of acceptance.

Reviewer #3:

I appreciate the authors inclusion of new data using C41S knock-in M17 cells to support target engagement with SP11 but the major critique regarding selectivity across the proteome from cellular studies is still not addressed. I believe this key question must be addressed since many of the claims are based on the ability of SP11 to selectively block covalent dimerization and activation for investigating Fis1 function. Unfortunately, I am not able to support publication till this major critique has been properly resolved.

Ans: The concern of the reviewer is that SP11 blocks H₂O₂-induced excessive mitochondrial fission by affecting proteins other than Fis1 and we agree that this should be explicitly addressed. We now added the following comments to address this concern (also added to the discussion section):

“SP11 includes a Cys reactive moiety and therefore there is a formal possibility that H₂O₂-induced mitochondrial fission is inhibited by interaction of SP11 with Cys-containing proteins other than Fis1. However, this possibility is unlikely, since SP11 does not affect mitochondrial fragmentation or ROS production in cells expressing Cys41Ser Fis1 (lack of an effect on MitoSOX in Cys41Ser knock-in M17 cells in the previous version and new data below showing mitochondrial morphology in Cys41Ser transfected MEFs; new Fig. 7g). Importantly, our data show that binding of SP11 to any other Cys (if any) in a Cys41Ser homozygous Fis1 cell is insufficient to reduce mitochondrial fragmentation or ROS production. SP11 does not bind Cys41Ser Fis1 nor does it bind Fis1 Cys41Ser Val56Cys double mutant (Fig. 5c). Finally, in addition to Cys41, Tyr38 is essential for SP11 binding to Fis1; Tyr38Glu does not bind SP11 (Fig. 6e), indicating further selectivity of the SP11 to its binding site on Fis1. Together, although the reactive Cys moiety in SP11 may interact with free Cys residues in other proteins, SP11 inhibits H₂O₂-

induced excessive mitochondrial fission in cells expressing WT Fis1 by inhibiting Fis1-mediated Drp1 binding.”

References

1. Nolden, K.A., Harwig, M.C. & Hill, R.B. Human Fis1 directly interacts with Drp1 in an evolutionarily conserved manner to promote mitochondrial fission. *J Biol Chem* **299**, 105380 (2023).
2. Ihenacho, U.K., Toro, R., Mansour, R.H. & Hill, R.B. A conserved, noncanonical insert in FIS1 mediates TBC1D15 and DRP1 recruitment for mitochondrial fission. *J Biol Chem* **299**, 105303 (2023).
3. Wells, R.C., Picton, L.K., Williams, S.C.P., Tan, F.J. & Hill, R.B. Direct binding of the dynamin-like GTPase, Dnm1, to mitochondrial dynamics protein Fis1 is negatively regulated by the Fis1 N-terminal arm. *J Biol Chem* **282**, 33769-33775 (2007).
4. Egner, J.M. et al. Structural studies of human fission protein FIS1 reveal a dynamic region important for GTPase DRP1 recruitment and mitochondrial fission. *J Biol Chem* **298**, 102620 (2022).